# RECURRENT HIERARCHICAL TOPIC-GUIDED NEURAL LANGUAGE MODELS

## ABSTRACT

To simultaneously capture syntax and global semantics from a text corpus, we propose a new larger-context recurrent neural network (RNN) based language model, which extracts recurrent hierarchical semantic structure via a dynamic deep topic model to guide natural language generation. Moving beyond a conventional RNN based language model that ignores long-range word dependencies and sentence order, the proposed model captures not only intra-sentence word dependencies, but also temporal transitions between sentences and inter-sentence topic dependences. For inference, we develop a hybrid of stochastic-gradient MCMC and recurrent autoencoding variational Bayes. Experimental results on a variety of real-world text corpora demonstrate that the proposed model not only outperforms state-of-the-art larger-context RNN-based language models, but also learns interpretable recurrent multilayer topics and generates diverse sentences and paragraphs that are syntactically correct and semantically coherent.

## 1 INTRODUCTION

Both topic and language models are widely used for text analysis. Topic models, such as latent Dirichlet allocation (LDA) (Blei et al., 2003; Griffiths & Steyvers, 2004; Hoffman et al., 2013) and its nonparametric Bayesian generalizations (Teh et al., 2006; Zhou & Carin, 2015), are well suited to extract document-level word concurrence patterns into latent topics from a text corpus. Their modeling power has been further enhanced by introducing multilayer deep representation (Srivastava et al., 2013; Mnih & Gregor, 2014; Gan et al., 2015; Zhou et al., 2016; Zhao et al., 2018; Zhang et al., 2018). While having semantically meaningful latent representation, they typically treat each document as a bag of words (BoW), ignoring word order (Griffiths et al., 2004; Wallach, 2006). Language models have become key components of various natural language processing (NLP) tasks, such as text summarization (Rush et al., 2015; Gehrmann et al., 2018), speech recognition (Mikolov et al., 2010; Graves et al., 2013), machine translation (Sutskever et al., 2014; Cho et al., 2014), and image captioning (Vinyals et al., 2015; Mao et al., 2015; Xu et al., 2015; Gan et al., 2017; Rennie et al., 2017). The primary purpose of a language model is to capture the distribution of a word sequence, commonly with a recurrent neural network (RNN) (Mikolov et al., 2011; Graves, 2013) or a Transformer based neural network (Vaswani et al., 2017; Dai et al., 2019; Devlin et al., 2019; Radford et al., 2018; 2019). In this paper, we focus on improving RNN-based language models that often have much fewer parameters and are easier to perform end-to-end training.

While RNN-based language models do not ignore word order, they often assume that the sentences of a document are independent to each other. This simplifies the modeling task to independently assigning probabilities to individual sentences, ignoring their orders and document context (Tian & Cho, 2016). Such language models may consequently fail to capture the long-range dependencies and global semantic meaning of a document (Dieng et al., 2017; Wang et al., 2018). To relax the sentence independence assumption in language modeling, Tian & Cho (2016) propose larger-context language models that model the context of a sentence by representing its preceding sentences as either a single or a sequence of BoW vectors, which are then fed directly into the sentence modeling RNN. An alternative approach attracting significant recent interest is leveraging topic models to improve RNN-based language models. Mikolov & Zweig (2012) use pre-trained topic model features as an additional input to the RNN hidden states and/or output. Dieng et al. (2017); Ahn et al. (2017) combine the predicted word distributions, given by both a topic model and a language model, under variational autoencoder (Kingma & Welling, 2013). Lau et al. (2017) introduce an attention based

convolutional neural network to extract semantic topics, which are used to extend the RNN cell. Wang et al. (2018) learn the global semantic coherence of a document via a neural topic model and use the learned latent topics to build a mixture-of-experts language model. Wang et al. (2019) further specify a Gaussian mixture model as the prior of the latent code in variational autoencoder, where each mixture component corresponds to a topic.

While clearly improving the performance of the end task, these existing topic-guided methods still have clear limitations. For example, they only utilize shallow topic models with only a single stochastic hidden layer in their data generation process. Note several neural topic models use deep neural networks to construct their variational encoders, but still use shallow generative models (decoders) (Miao et al., 2017; Srivastava & Sutton, 2017). Another key limitation lies in ignoring the sentence order, as they treat each document as a bag of sentences. Thus once the topic weight vector learned from the document context is given, the task is often reduced to independently assigning probabilities to individual sentences (Lau et al., 2017; Wang et al., 2018; 2019).

In this paper, as depicted in Fig. 1, we propose to use recurrent gamma belief network (rGBN) to guide a stacked RNN for language modeling. We refer to the model as rGBN-RNN, which integrates rGBN (Guo et al., 2018), a deep recurrent topic model, and stacked RNN (Graves, 2013; Chung et al., 2017), a neural language model, into a novel larger-context RNN-based language model. It simultaneously learns a deep recurrent topic model, extracting document-level multi-layer word concurrence patterns and sequential topic weight vectors for sentences, and an expressive language model, capturing both short- and long-range word sequential dependencies. For inference, we equip rGBN-RNN (decoder) with a novel variational recurrent inference network (encoder), and train it end-to-end by maximizing the evidence lower bound (ELBO). Different from the stacked RNN based language model in Chung et al. (2017), which relies on three types of customized training operations (UPDATE, COPY, FLUSH) to extract multi-scale structures, the language model in rGBN-RNN learns such structures purely under the guidance of the temporally and hierarchically connected stochastic layers of rGBN. The effectiveness of rGBN-RNN as a new larger-context language model is demonstrated both quantitatively, with perplexity and BLEU scores, and qualitatively, with interpretable latent structures and randomly generated sentences and paragraphs. Notably, rGBN-RNN can generate a paragraph consisting of a sequence of semantically coherent sentences.

## 2 RECURRENT HIERARCHICAL TOPIC-GUIDED LANGUAGE MODEL

Denote a document of $J$ sentences as $\mathcal{D} = (S_1, S_2, \ldots, S_J)$, where $S_j = (y_{j,1}, \ldots, y_{j,T_j})$ consists of $T_j$ words from a vocabulary of size $V$. Conventional statistical language models often only focus on the word sequence within a sentence. Assuming that the sentences of a document are independent to each other, they often define $P(\mathcal{D}) \approx \prod_{j=1}^{J} P(S_j) = \prod_{j=1}^{J} \prod_{t=2}^{T_j} p(y_{j,t} \mid y_{j,<t}) p(y_{j,1})$. RNN based neural language models define the conditional probability of each word $y_{j,t}$ given all the previous words $y_{j,<t}$ within the sentence $S_j$, through the softmax function of a hidden state $\boldsymbol{h}_{j,t}$, as

$$p(y_{j,t} \mid y_{j,<t}) = p(y_{j,t} \mid \boldsymbol{h}_{j,t}), \quad \boldsymbol{h}_{j,t} = f(\boldsymbol{h}_{j,<t}, y_{j,t-1}), \tag{1}$$

where $f(\cdot)$ is a non-linear function typically defined as an RNN cell, such as long short-term memory (LSTM) (Hochreiter & Schmidhuber, 1997) and gated recurrent unit (GRU) (Cho et al., 2014).

These RNN-based statistical language models are typically applied only at the word level, without exploiting the document context, and hence often fail to capture long-range dependencies. While Dieng et al. (2017); Lau et al. (2017); Wang et al. (2018; 2019) remedy the issue by guiding the language model with a topic model, they still treat a document as a bag of sentences, ignoring the order of sentences, and lack the ability to extract hierarchical and recurrent topic structures.

We introduce rGBN-RNN, as depicted in Fig. 1(a), as a new larger-context language model. It consists of two key components: (i) a hierarchical recurrent topic model (rGBN), and (ii) a stacked RNN based language model. We use rGBN to capture both global semantics across documents and long-range inter-sentence dependencies within a document, and use the language model to learn the local syntactic relationships between the words within a sentence. Similar to Lau et al. (2017); Wang et al. (2018), we represent a document as a sequence of sentence-context pairs as $(\{S_1, \boldsymbol{d}_1\}, \ldots, \{S_J, \boldsymbol{d}_J\})$, where $\boldsymbol{d}_j \in \mathbb{Z}_+^{V_c}$ summarizes the document excluding $S_j$, specifically $(S_1, \ldots, S_{j-1}, S_{j+1}, \ldots, S_J)$, into a BoW count vector, with $V_c$ as the size of the vocabulary excluding stop words. Note a naive way is to treat each sentence as a document, use a dynamic topic model

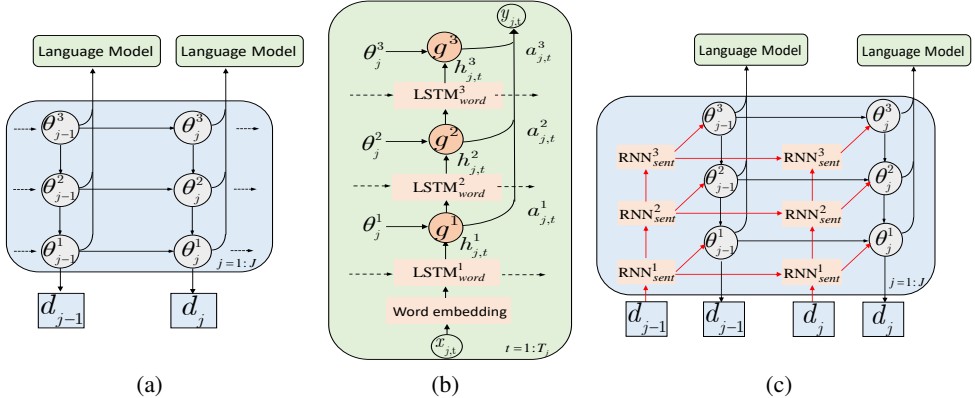

(a)  (b)  (c)

Figure 1: (a) The generative model of a three-hidden-layer rGBN-RNN, where the bottom part is the deep recurrent topic model (rGBN), document contexts of consecutive sentences are used as observed data, and upper is the language model. (b) Overview of the language model component, where input $x_{j,t}$ denotes the $t$th word in $j$th sentence of a document, $x_{j,t} = y_{j,t-1}$, $\boldsymbol{h}_{j,t}^l$ is the hidden state of the stacked RNN at time step $t$, and $\boldsymbol{\theta}_j^l$ is the topic weight vector of sentence $j$ at layer $l$. (c) The overall architecture of the proposed model, including the decoder (rGBN and language model) and encoder (variational recurrent inference), where the red arrows denote the inference of latent topic weight vectors, black ones the data generation.

(Blei & Lafferty, 2006) to capture the temporal dependencies of the latent topic-weight vectors, which is fed to the RNN to model the word sequence of the corresponding sentence. However, the sentences are often too short to be well modeled by a topic model. In our setting, as $\boldsymbol{d}_j$ summarizes the document-level context of $S_j$, it is in general sufficiently long for topic modeling. Note during testing, we redefine $\boldsymbol{d}_j$ as the BoW vector summarizing only the preceding sentences, *i.e.*, $S_{1:j-1}$, which will be further clarified when presenting experimental results.

## 2.1 HIERARCHICAL RECURRENT TOPIC MODEL

Shown in Fig. 1 (a), to model the time-varying sentence-context count vectors $\boldsymbol{d}_j$ in document $\mathcal{D}$, the generative process of the rGBN component, from the top to bottom hidden layers, is expressed as

$$\boldsymbol{\theta}_j^L \sim \text{Gam}\left(\boldsymbol{\Pi}^L \boldsymbol{\theta}_{j-1}^L, \, \tau_0\right), \cdots, \, \boldsymbol{\theta}_j^l \sim \text{Gam}\left(\boldsymbol{\Phi}^{l+1} \boldsymbol{\theta}_j^{l+1} + \boldsymbol{\Pi}^l \boldsymbol{\theta}_{j-1}^l, \, \tau_0\right), \cdots,$$
$$\boldsymbol{\theta}_j^1 \sim \text{Gam}\left(\boldsymbol{\Phi}^2 \boldsymbol{\theta}_j^2 + \boldsymbol{\Pi}^1 \boldsymbol{\theta}_{j-1}^1, \, \tau_0\right), \, \boldsymbol{d}_j \sim \text{Pois}\left(\boldsymbol{\Phi}^1 \boldsymbol{\theta}_j^1\right), \tag{2}$$

where $\boldsymbol{\theta}_j^l \in \mathbb{R}_+^{K_l}$ denotes the gamma distributed topic weight vectors of sentence $j$ at layer $l$, $\boldsymbol{\Pi}^l \in \mathbb{R}_+^{K_l \times K_l}$ the transition matrix of layer $l$ that captures cross-topic temporal dependencies, $\boldsymbol{\Phi}^l \in \mathbb{R}_+^{K_{l-1} \times K_l}$ the loading matrix at layer $l$, $K_l$ the number of topics of layer $l$, and $\tau_0 \in \mathbb{R}_+$ a scaling hyperparameter. At $j = 1$, $\boldsymbol{\theta}_1^l \sim \text{Gam}\left(\boldsymbol{\Phi}^{l+1} \boldsymbol{\theta}_1^{l+1}, \tau_0\right)$ for $l = 1, \ldots, L - 1$ and $\boldsymbol{\theta}_1^L \sim \text{Gam}\left(\nu, \tau_0\right)$, where $\nu = \mathbf{1}_{K_L}$. Finally, Dirichlet priors are placed on the columns of $\boldsymbol{\Pi}^l$ and $\boldsymbol{\Phi}^l$, *i.e.*, $\boldsymbol{\pi}_k^l$ and $\boldsymbol{\phi}_k^l$, which not only makes the latent representation more identifiable and interpretable, but also facilitates inference. The count vector $\boldsymbol{d}_j$ can be factorized into the product of $\boldsymbol{\Phi}^1$ and $\boldsymbol{\theta}_j^1$ under the Poisson likelihood. The shape parameters of $\boldsymbol{\theta}_j^l \in \mathbb{R}_+^{K_l}$ can be factorized into the sum of $\boldsymbol{\Phi}^{l+1} \boldsymbol{\theta}_j^{l+1}$, capturing inter-layer hierarchical dependence, and $\boldsymbol{\Pi}^l \boldsymbol{\theta}_{j-1}^l$, capturing intra-layer temporal dependence. rGBN not only captures the document-level word occurrence patterns inside the training text corpus, but also the sequential dependencies of the sentences inside a document. Note ignoring the recurrent structure, rGBN will reduce to the gamma belief network (GBN) of Zhou et al. (2016), which can be considered as a multi-stochastic-layer deep generalization of LDA (Cong et al., 2017a). If ignoring its hierarchical structure (i.e., $L = 1$), rGBN reduces to Poisson–gamma dynamical systems (Schein et al., 2016). We refer to the rGBN-RNN without its recurrent structure as GBN-RNN, which no longer models sequential sentence dependencies; see Appendix A for more details.

## 2.2 LANGUAGE MODEL

Different from a conventional RNN-based language model, which predicts the next word only using the preceding words within the sentence, we integrate the hierarchical recurrent topic weight vectors $\boldsymbol{\theta}_j^l$ into the language model to predict the word sequence in the $j$th sentence. Our proposed language model is built upon the stacked RNN proposed in Graves (2013); Chung et al. (2017), but with the help of rGBN, it no longer requires specialized training heuristics to extract multi-scale structures. As shown in Fig. 1 (b), to generate $y_{j,t}$, the $t^{\text{th}}$ token of sentence $j$ in a document, we construct the hidden states $\boldsymbol{h}_{j,t}^l$ of the language model, from the bottom to top layers, as

$$\boldsymbol{h}_{j,t}^l = \left\{ \begin{array}{ll} \text{LSTM}_{\text{word}}^l\left(\boldsymbol{h}_{j,t-1}^l, \boldsymbol{W}_e\left[x_{j,t}\right]\right), & \text{if } l = 1, \\ \text{LSTM}_{\text{word}}^l\left(\boldsymbol{h}_{j,t-1}^l, \boldsymbol{a}_{j,t}^{l-1}\right), & \text{if } L \geq l > 1, \end{array} \right. \tag{3}$$

where $\text{LSTM}_{\text{word}}^l$ denotes the word-level LSTM at layer $l$, $\mathbf{W}_e \in \mathbb{R}^V$ are word embeddings to be learned, and $x_{j,t} = y_{j,t-1}$. Note $\boldsymbol{a}_{j,t}^l$ denotes the coupling vector, which combines the temporal topic weight vectors $\boldsymbol{\theta}_j^l$ and hidden output of the word-level LSTM $\boldsymbol{h}_{j,t-1}^l$ at each time step $t$.

Following Lau et al. (2017), a gating unit similar to GRU (Cho et al., 2014) combines $\boldsymbol{\theta}_j^l$ of sentence $j$ with its hidden state $\boldsymbol{h}_{j,t}^l$ of word-level LSTM at layer $l$ and time $t$ as $\boldsymbol{a}_{j,t}^l = g^l\left(\boldsymbol{h}_{j,t}^l, \boldsymbol{\theta}_j^l\right)$. We defer the details on $g^l$ to Appendix B. Denote $\mathbf{W}_o$ as a weight matrix with $V$ rows and $\boldsymbol{a}_{j,t}^{1:L}$ as the concatenation of $\boldsymbol{a}_{j,t}^l$ across all layers; different from (1), the conditional probability of $y_{j,t}$ becomes

$$p\left(y_{j,t} \mid y_{j,<t}, \boldsymbol{\theta}_j^l\right) = \text{softmax}\left(\mathbf{W}_o \boldsymbol{a}_{j,t}^{1:L}\right). \tag{4}$$

There are two main reasons for combining all the latent representations $\boldsymbol{a}_{j,t}^{1:L}$ for language modeling. First, the latent representations exhibit different statistical properties at different stochastic layers of rGBN-RNN, and hence are combined together to enhance their representation power. Second, having "skip connections" from all hidden layers to the output one makes it easier to train the proposed network, reducing the number of processing steps between the bottom of the network and the top and hence mitigating the "vanishing gradient" problem (Graves, 2013).

To sum up, as depicted in Fig. 1 (a), the topic weight vector $\boldsymbol{\theta}_j^l$ of sentence $j$ quantifies the topic usage of its document context $\boldsymbol{d}_j$ at layer $l$. It is further used as an additional feature of the language model to guide the word generation inside sentence $j$, as shown in Fig. 1 (b). It is clear that rGBN-RNN has two temporal structures: a deep recurrent topic model to extract the temporal topic weight vectors from the sequential document contexts, and a language model to estimate the probability of each sentence given its corresponding hierarchical topic weight vector. Characterizing the word-sentence-document hierarchy to incorporate both intra- and inter-sentence information, rGBN-RNN learns more coherent and interpretable topics and increases the generative power of the language model. Distinct from existing topic-guided language models, the temporally related hierarchical topics of rGBN exhibit different statistical properties across layers, which better guides language model to improve its language generation ability.

## 2.3 MODEL LIKELIHOOD AND INFERENCE

For rGBN-RNN, given $\{\boldsymbol{\Phi}^l, \boldsymbol{\Pi}^l\}_{l=1}^L$, the marginal likelihood of the sequence of sentence-context pairs $(\{\boldsymbol{s}_1, \boldsymbol{d}_1\}, \ldots, \{\boldsymbol{s}_J, \boldsymbol{d}_J\})$ of document $\mathcal{D}$ is defined as

$$P\left(\mathcal{D} \mid \{\boldsymbol{\Phi}^l, \boldsymbol{\Pi}^l\}_{l=1}^L\right) = \int \prod_{j=1}^J \left\{ p\left(\boldsymbol{d}_j \mid \boldsymbol{\Phi}^1 \boldsymbol{\theta}_j^1\right) \left[\prod_{t=1}^{T_j} p\left(y_{jt} \mid y_{j,<t}, \boldsymbol{\theta}_j^{1:L}\right)\right] \left[\prod_{l=1}^L p\left(\boldsymbol{\theta}_j^l \mid e_j^l, \tau_0\right)\right] \right\} d\boldsymbol{\theta}_{1:J}^{1:L}, \tag{5}$$

where $e_j^l := \boldsymbol{\Phi}^{l+1} \boldsymbol{\theta}_j^{l+1} + \boldsymbol{\Pi}^l \boldsymbol{\theta}_{j-1}^l$. The inference task is to learn the parameters of both the topic model and language model components. One naive solution is to alternate the training between these two components in each iteration: First, the topic model is trained using a sampling based iterative algorithm provided in Guo et al. (2018); Second, the language model is trained with maximum likelihood estimation under a standard cross-entropy loss. While this naive solution can utilize readily available inference algorithms for both rGBN and the language model, it may suffer from stability and convergence issues. Moreover, the need to perform a sampling based iterative algorithm for rGBN inside each iteration limits the scalability of the model for both training and testing.

---

**Algorithm 1** Hybrid SG-MCMC and recurrent autoencoding variational inference for rGBN-RNN.

---
Set mini-batch size $m$ and the number of layer $L$

Initialize encoder and neural language model parameter parameter $\mathbf{\Omega}$, and topic model parameter $\{\mathbf{\Phi}^l, \mathbf{\Pi}^l\}_{l=1}^L$.

**for** $iter = 1, 2, \cdots$ **do**

    Randomly select a mini-batch of $m$ documents consisting of $J$ sentences to form a subset $\mathbf{X} = \{\boldsymbol{d}_{i,1:J}, \boldsymbol{s}_{i,1:J}\}_{i=1}^m$;

    Draw random noise $\{\boldsymbol{\epsilon}_{i,j}^l\}_{i=1,j=1,l=1}^{m,J,L}$ from uniform distribution;

    Calculate $\nabla_{\mathbf{\Omega}} L\left(\mathbf{\Omega}, \mathbf{\Phi}^l, \mathbf{\Pi}^l; \mathbf{X}, \boldsymbol{\epsilon}_{i,j}^l\right)$ according to (6), and update $\mathbf{\Omega}$; Sample $\boldsymbol{\theta}_{i,j}^l$ from (7) and (8) via $\mathbf{\Omega}$ to update $\{\mathbf{\Pi}^l\}_{l=1}^L$ and $\{\mathbf{\Phi}^l\}_{l=1}^L$, will be described in Appendix C;

**end for**

---

To this end, we introduce a variational recurrent inference network (encoder) to learn the latent temporal topic weight vectors $\boldsymbol{\theta}_{1:J}^{1:L}$. Denoting $Q = \prod_{j=1}^J \prod_{l=1}^L q(\boldsymbol{\theta}_j^l \mid \boldsymbol{d}_{\leq j})$, the ELBO of the log marginal likelihood shown in (5) can be constructed as

$$L = \sum_{j=1}^J \mathbb{E}_Q \left[\ln p\left(\boldsymbol{d}_j \mid \mathbf{\Phi}^1 \boldsymbol{\theta}_j^1\right) + \sum_{t=1}^{T_j} \ln p\left(y_{j,t} \mid y_{j,<t}, \boldsymbol{\theta}_j^{1:L}\right)\right] - \sum_{j=1}^J \sum_{l=1}^L \mathbb{E}_Q \left[\ln \frac{q\left(\boldsymbol{\theta}_j^l \mid \boldsymbol{d}_{\leq j}\right)}{p\left(\boldsymbol{\theta}_j^l \mid \boldsymbol{e}_j^l, \tau_0\right)}\right], \quad (6)$$

which unites both the terms that are primarily responsible for training the recurrent hierarchical topic model component, and terms for training the neural language model component. Similar to Zhang et al. (2018), we define $q(\boldsymbol{\theta}_j^l \mid \boldsymbol{d}_{\leq j}) = \text{Weibull}(\boldsymbol{k}_j^l, \boldsymbol{\lambda}_j^l)$, a random sample from which can be obtained by transforming standard uniform variables $\boldsymbol{\epsilon}_j^l$ as

$$\boldsymbol{\theta}_j^l = \boldsymbol{\lambda}_j^l\left(-\ln(1 - \boldsymbol{\epsilon}_j^l)\right)^{1/\boldsymbol{k}_j^l}. \quad (7)$$

To capture the temporal dependencies between the topic weight vectors, both $\boldsymbol{k}_j^l$ and $\boldsymbol{\lambda}_j^l$, from the bottom to top layers, can be expressed as

$$\boldsymbol{h}_j^{s,l} = \text{RNN}_{\text{sent}}^l\left(\boldsymbol{h}_{j-1}^{s,l}, \boldsymbol{h}_j^{s,l-1}\right), \quad \boldsymbol{k}_j^l = f_{\boldsymbol{k}}^l\left(\boldsymbol{h}_j^{s,l}\right), \boldsymbol{\lambda}_j^l = f_{\boldsymbol{\lambda}}^l\left(\boldsymbol{h}_j^{s,l}\right), \quad (8)$$

where $\boldsymbol{h}_j^{s,0} = \boldsymbol{d}_j$, $\boldsymbol{h}_0^{s,l} = 0$, $\text{RNN}_{\text{sent}}^l$ denotes the sentence-level recurrent encoder at layer $l$ implemented with a basic RNN cell, capturing the sequential relationship between sentences within a document, $\boldsymbol{h}_j^{s,l}$ denotes the hidden state of $\text{RNN}_{\text{sent}}^l$, and superscript $s$ in $\boldsymbol{h}_j^{s,l}$ denotes "sentence-level RNN" used to distinguish the hidden state of language model in (3) . Note both $f_{\boldsymbol{k}}^l$ and $f_{\boldsymbol{\lambda}}^l$ are nonlinear functions mapping state $\boldsymbol{h}_j^{s,l}$ to the parameters of $\boldsymbol{\theta}_j^l$, implemented with $f(\boldsymbol{x}) = \ln(1 + \exp(\mathbf{W}\boldsymbol{x} + \boldsymbol{b}))$.

Rather than finding a point estimate of the global parameters $\{\mathbf{\Phi}^l, \mathbf{\Pi}^l\}_{l=1}^L$ of the rGBN, we adopt a hybrid inference algorithm by combining TLASGR-MCMC described in Cong et al. (2017a); Zhang et al. (2018) and our proposed recurrent variational inference network. In other words, the global parameters $\{\mathbf{\Phi}^l, \mathbf{\Pi}^l\}_{l=1}^L$ can be sampled with TLASGR-MCMC, while the parameters of the language model and variational recurrent inference network, denoted by $\mathbf{\Omega}$, can be updated via stochastic gradient descent (SGD) by maximizing the ELBO in (6). We describe a hybrid variational/sampling inference for rGBN-RNN in Algorithm 1 and provide more details about sampling $\{\mathbf{\Phi}^l, \mathbf{\Pi}^l\}_{l=1}^L$ with TLASGR-MCMC in Appendix C. We defer the details on model complexity to Appendix E.

To sum up, as shown in Fig. 1(c), the proposed rGBN-RNN works with a recurrent variational autoencoder inference framework, which takes the document context of the $j$th sentence within a document as input and learns hierarchical topic weight vectors $\boldsymbol{\theta}_j^{1:L}$ that evolve sequentially with $j$. The learned topic vectors in different layer are then used to reconstruct the document context input and as an additional feature for the language model to generate the $j$th sentence.

## 3   Experimental results

We consider three publicly available corpora, including APNEWS, IMDB, and BNC. The links, preprocessing steps, and summary statistics for them are deferred to Appendix D. We consider a recurrent variational inference network for rGBN-RNN to infer $\boldsymbol{\theta}_j^l$, as shown in Fig. 1(c), whose number of hidden units in (8) are set the same as the number of topics in the corresponding layer.

Following Lau et al. (2017), word embeddings are pre-trained 300-dimension word2vec Google News vectors (https://code.google.com/archive/p/word2vec/). Dropout with a rate of $0.4$ is used to the input of the stacked-RNN at each layer, i.e., $\boldsymbol{a}_{j,t}^l$ or $\boldsymbol{W_e}\left[x_{j,t}\right]$ in (3). The gradients are clipped if the norm of the parameter vector exceeds 5. We use the Adam optimizer (Kingma & Ba, 2015) with learning rate $10^{-3}$. The length of an input sentence is fixed to 30. We set the mini-batch size as $8$, number of training epochs as 5, and scaling hyperparameter $\tau_0$ as 1. Code in TensorFlow is provided.

## 3.1 QUANTITATIVE COMPARISON

**Perplexity:** For fair comparison, we use standard language model perplexity as the evaluation metric, by considering the following baselines: *(i)* a standard LSTM language model (Hochreiter & Schmidhuber, 1997); *(ii)* LCLM (Tian & Cho, 2016), a larger-context language model that incorporates context from preceding sentences, which are treated as a bag of words; *(iii)* a standard LSTM language model incorporating the topic information of a separately trained LDA (LDA+LSTM); *(iv)* Topic-RNN (Dieng et al., 2017), a hybrid model rescoring the prediction of the next word by incorporating the topic information through a linear transformation; *(v)* TDLM (Lau et al., 2017), a joint learning framework which learns a convolutional based topic model and a language model simultaneously. *(vi)* TCNLM (Wang et al., 2018), which extracts the global semantic coherence of a document via a neural topic model, with the probability of each learned latent topic further adopted to build a mixture-of-experts language model; *(vii)* TGVAE (Wang et al., 2019), combining a variational auto-encoder based neural sequence model with a neural topic model; *(viii)* GBN-RNN, a simplified rGBN-RNN that removes the recurrent structure of its rGBN component.

For rGBN-RNN, to ensure the information about the words in the $j$th sentence to be predicted is not leaking through the sequential document context vectors at the testing stage, the input $\boldsymbol{d}_j$ in (8) only summarizes the preceding sentences $S_{<j}$. For GBN-RNN, following TDLM (Lau et al., 2017) and TCNLM (Wang et al., 2018), all the sentences in a document, excluding the one being predicted, are used to obtain the BoW document context. As shown in Table 1, rGBN-RNN outperforms all baselines, and the trend of improvement continues as its number of layers increases, indicating the effectiveness of assimilating recurrent hierarchical topic information. rGBN-RNN consistently outperforms GBN-RNN, suggesting the benefits of exploiting the sequential dependencies of the sentence-contexts for language modeling. Moreover, comparing Table 1 and Table 4 of Appendix E suggests rGBN-RNN, with its hierarchical and temporal topical guidance, achieves better performance with fewer parameters than comparable RNN-based baselines.

Note that for language modeling, there has been significant recent interest in replacing RNNs with Transformer (Vaswani et al., 2017), which consists of stacked multi-head self-attention modules, and its variants (Dai et al., 2019; Devlin et al., 2019; Radford et al., 2018; 2019). While Transformer based language models have been shown to be powerful in various natural language processing tasks, they often have significantly more parameters, require much more training data, and take much longer to train than RNN-based language models. For example, Transformer-XL with 12L and that with 24L (Dai et al., 2019), which improve Transformer to capture longer-range dependencies, have 41M and 277M parameters, respectively, while the proposed rGBN-RNN with three stochastic hidden layers has as few as 7.3M parameters, as shown in Table 4, when used for language modeling. From a structural point-of-view, we consider the proposed rGBN-RNN as complementary to rather than competing with Transformer based language models, and consider replacing RNN with Transformer to construct rGBN guided Transformer as a promising future extension.

**BLEU:** Following Wang et al. (2019), we use test-BLEU to evaluate the quality of generated sentences with a set of real test sentences as the reference, and self-BLEU to evaluate the diversity of the generated sentences (Zhu et al., 2018). Given the global parameters of the deep recurrent topic model (rGBN) and language model, we can generate the sentences by following the data generation process of rGBN-RNN: we first generate topic weight vectors $\boldsymbol{\theta}_j^L$ randomly and then downward propagate it through the rGBN as in (2) to generate $\boldsymbol{\theta}_j^{<L}$. By assimilating the random draw topic weight vectors with the hidden states of the language model in each layer depicted in (3), we generate a corresponding sentence, where we start from a zero hidden state at each layer in the language model, and sample words sequentially until the end-of-the-sentence symbol is generated. Comparisons of the BLEU scores between different methods are shown in Fig. 2, using the benchmark tool in Texygen (Zhu et al., 2018); We show below BLEU-3 and BLEU-4 for BNC and defer the analogous plots for IMDB and APNEWS to Appendix G and H. Note we set the validation dataset as the ground-truth.

Table 1: Comparison of perplexity on three different datasets.

| Model | LSTM Size | Topic Size | Perplexity | | |
|---|---|---|---|---|---|
| | | | APNEWS | IMDB | BNC |
| LCLM (Tian & Cho, 2016) | 600 | — | 54.18 | 67.78 | 96.50 |
| | 900-900 | — | 50.63 | 67.86 | 87.77 |
| LDA+LSTM (Lau et al., 2017) | 600 | 100 | 55.52 | 69.64 | 96.50 |
| | 900-900 | 100 | 50.75 | 63.04 | 87.77 |
| TopicRNN (Dieng et al., 2017) | 600 | 100 | 54.54 | 67.83 | 93.57 |
| | 900-900 | 100 | 50.24 | 61.59 | 84.62 |
| TDLM (Lau et al., 2017) | 600 | 100 | 52.75 | 63.45 | 85.99 |
| | 900-900 | 100 | 48.97 | 59.04 | 81.83 |
| TCNLM (Wang et al., 2018) | 600 | 100 | 52.63 | 62.64 | 86.44 |
| | 900-900 | 100 | 47.81 | 56.38 | 80.14 |
| TGVAE (Wang et al., 2019) | 600 | 50 | 48.73 | 57.11 | 87.86 |
| basic-LSTM (Hochreiter & Schmidhuber, 1997) | 600 | — | 64.13 | 72.14 | 102.89 |
| | 900-900 | — | 58.89 | 66.47 | 94.23 |
| | 900-900-900 | — | 60.13 | 65.16 | 95.73 |
| GBN-RNN | 600 | 100 | 47.42 | 57.01 | 86.39 |
| | 600-512 | 100-80 | 44.64 | 55.42 | 82.95 |
| | 600-512-256 | 100-80-50 | 44.35 | 54.53 | 80.25 |
| rGBN-RNN | 600 | 100 | 46.35 | 55.76 | 81.94 |
| | 600-512 | 100-80 | 43.26 | 53.82 | 80.25 |
| | 600-512-256 | 100-80-50 | **42.71** | **51.36** | **79.13** |

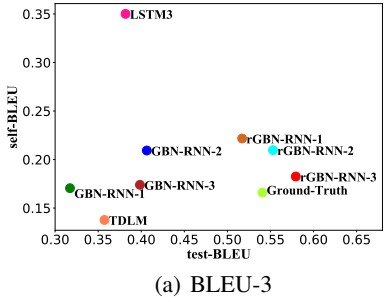

(a) BLEU-3

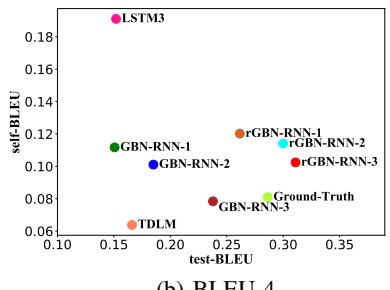

(b) BLEU-4

Figure 2: BLEU scores of different methods for BNC. x-axis denotes test-BLEU, y-axis self-BLEU, and a better BLEU score would fall within the lower right corner.

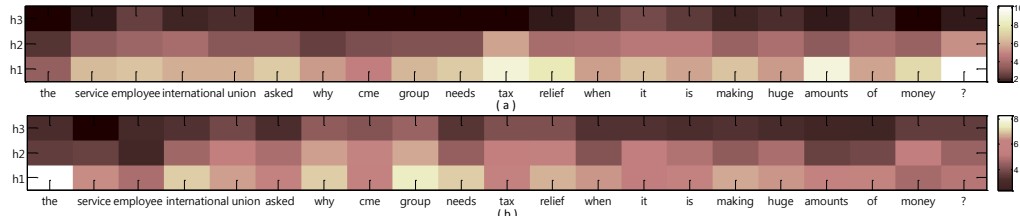

Figure 3: Visualization of the $L_2$-norm of the hidden states of the language model of rGBN-RNN, shown in the top-row, and that of GBN-RNN, shown in the bottom row.

For all datasets, it is clear that rGBN-RNN yields both higher test-BLEU and lower self-BLEU scores than related methods do, indicating the stacked-RNN based language model in rGBN-RNN generalizes well and does not suffer from mode collapse (i.e., low diversity).

## 3.2 QUALITATIVE ANALYSIS

**Hierarchical structure of language model:** In Fig. 3, we visualize the hierarchical multi-scale structures learned with the language model of rGBN-RNN and that of GBN-RNN, by visualizing the $L_2$-norm of the hidden states in each layer, while reading a sentence from the APNEWS validation set as "*the service employee international union asked why cme group needs tax relief when it is making huge amounts of money?*" As shown in Fig. 3(a), in the bottom hidden layer (h1), the $L_2$ norm sequence varies quickly from word to word, except within short phrases such "service employee", "international union," and "tax relief," suggesting layer h1 is in charge of capturing short-term local dependencies. By contrast, in the top hidden layer (h3), the $L_2$ norm sequence varies slowly and exhibits semantic/syntactic meaningful long segments, such as "service employee international union," "asked why cme group needs tax relief," "when it is," and "making huge amounts of," suggesting that layer h3 is in charge of capturing long-range dependencies. Therefore, the language model in

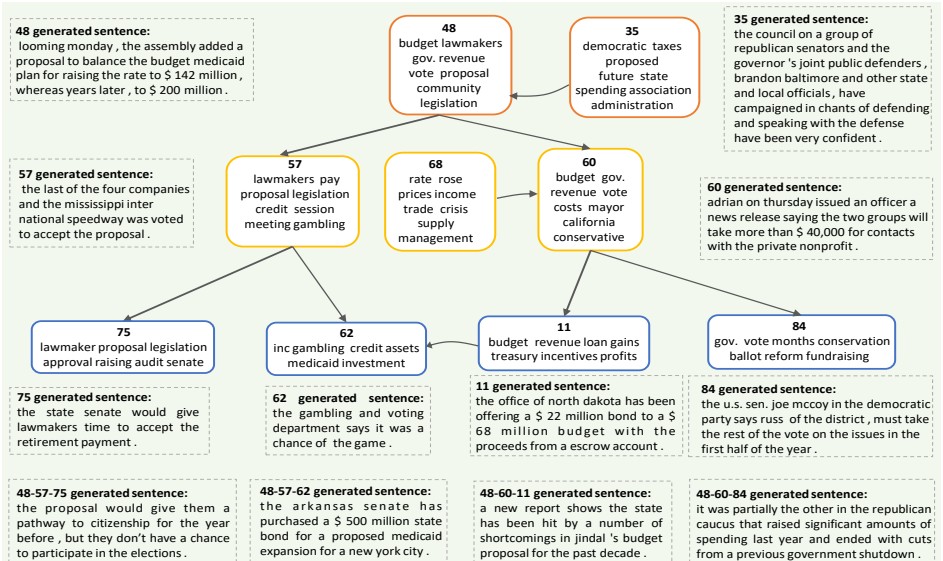

Figure 4: Topics and their temporal trajectories inferred by a three-hidden-layer rGBN-RNN from the APNEWS dataset, and the generated sentences under topic guidance (best viewed in color). Top words of each topic at layer 3, 2, 1 are shown in orange, yellow and blue boxes respectively, and each sentence is shown in a dotted line box labeled with the corresponding topic index. Sentences generated with a combination of topics in different layers are at the bottom of the figure.

rGBN-RNN can allow more specific information to transmit through lower layers, while allowing more general higher level information to transmit through higher layers. Our proposed model have the ability to learn hierarchical structure of the sequence, despite without designing the multiscale RNNs on purpose like Chung et al. (2017). We also visualize the language model of GBN-RNN in Fig. 3(b); with much less smoothly time-evolved deeper layers, GBN-RNN fails to utilize its stacked RNN structure as effectively as rGBN-RNN does. This suggests that the language model is much better trained in rGBN-RNN than in GBN-RNN for capturing long-range temporal dependencies, which helps explain why rGBN-RNN exhibits clearly boosted BLEU scores in comparison to GBN-RNN.

**Hierarchical topics:** We present an example topic hierarchy inferred by a three-layer rGBN-RNN from APNEWS. In Fig. 4, we select a large-weighted topic at the top hidden layer and move down the network to include any lower-layer topics connected to their ancestors with sufficiently large weights. Horizontal arrows link temporally related topics at the same layer, while top-down arrows link hierarchically related topics across layers. For example, topic 48 of layer 3 on "budget, lawmakers, gov., revenue," is related not only in hierarchy to topic 57 on "lawmakers, pay, proposal, legislation" and topic 60 of the lower layer on "budget, gov., revenue, vote, costs, mayor," but also in time to topic 35 of the same layer on "democratic, taxes, proposed, future, state." Highly interpretable hierarchical relationships between the topics at different layers, and temporal relationships between the topics at the same layer are captured by rGBN-RNN, and the topics are often quite specific semantically at the bottom layer while becoming increasingly more general when moving upwards.

**Sentence generation under topic guidance:** Given the learned rGBN-RNN, we can sample the sentences both conditioning on a single topic of a certain layer and on a combination of the topics from different layers. Shown in the dotted-line boxes in Fig. 4, most of the generated sentences conditioned on a single topic or a combination of topics are highly related to the given topics in terms of their semantical meanings but not necessarily in key words, indicating the language model is successfully guided by the recurrent hierarchical topics. These observations suggest that rGBN-RNN has successfully captured syntax and global semantics simultaneously for natural language generation.

**Sentence/paragraph generation conditioning on a paragraph:** Given the GBN-RNN and rGBN-RNN learned on APNEWS, we further present the generated sentences conditioning on a paragraph, as shown in Fig. 5. To randomly generate sentences, we encode the paragraph into a hierarchical latent representation and then feed it into the stacked-RNN. Besides, we can generate a paragraph with rGBN-RNN, using its recurrent inference network to encode the paragraph into a dynamic hierarchical latent representation, which is fed into the language model to predict the word sequence

**Document**
⦾ the senate sponsor (...) , a house committee last week removed photo ids issued by public colleges and universities from the measure sponsored by republican rep. susan lynn , who said she agreed with the change . the house approved the bill on a 65-30 vote on monday evening . but republican sen. bill ketron in a statement noted that the upper chamber overwhelmingly rejected efforts to take student ids out of the bill when it passed 21-8 earlier this month . ketron said he would take the bill to conference committee if needed .

**Generated Sentences with GBN-RNN**
⦾ if the house and senate agree , it will be the first time they 'll have to seek their first meeting .
⦾ the proposal would also give lawmakers with more money to protect public safety , he said .

**Generated Sentences with rGBN-RNN**
⦾ the proposal , which was introduced in the house on a vote on wednesday , has already passed the senate floor to the house .
⦾ the city commission voted last week to approve the law , which would have allowed the council to approve the new bill .

**Generated temporal Sentences with rGBN-RNN (Paragraph)**
⦾ senate president pro tem joe scarnati said the governor 's office has never resolved the deadline for a vote in the house . the proposal is a new measure version of the bill to enact a senate committee to approve the emergency manager 's emergency license . the house gave the bill to six weeks of testimony , but the vote now goes to the full house for consideration . jackson signed his paperwork wednesday with the legislature . the proposal would also give lawmakers with more money to protect public safety , he said . "a spokesman for the federal department of public safety says it has been selected for a special meeting for the state senate to investigate his proposed law . a new state house committee has voted to approve a measure to let idaho join a national plan to ban private school systems at public schools . the campaign also launched a website at the university of california , irvine , which are studying the current proposal .

Figure 5: Examples of generated sentences and paragraph conditioned on a document from APNEWS (green denotes novel words, blue the key words in document and generated sentences.)

in each sentence of the input paragraph. It is clear that both the proposed GBN-RNN and rGBN-RNN can successfully capture the key textual information of the input paragraph, and generate diverse realistic sentences. Interestingly, the rGBN-RNN can generate semantically coherent paragraphs, incorporating contextual information both within and beyond the sentences. Note that with the topics that extract the document-level word cooccurrence patterns, our proposed models can generate semantically-meaningful words, which may not exist in the original document.

## 4    CONCLUSION

We propose a recurrent gamma belief network (rGBN) guided neural language modeling framework, a novel method to learn a language model and a deep recurrent topic model simultaneously. For scalable inference, we develop hybrid SG-MCMC and recurrent autoencoding variational inference, allowing efficient end-to-end training. Experiments results conducted on real world corpora demonstrate that the proposed models outperform a variety of shallow-topic-model-guided neural language models, and effectively generate the sentences from the designated multi-level topics or noise, while inferring interpretable hierarchical latent topic structure of document and hierarchical multiscale structures of sequences. For future work, we plan to extend the proposed models to specific natural language processing tasks, such as machine translation, image paragraph captioning, and text summarization. Another promising extension is to replace the stacked-RNN in rGBN-RNN with Transformer, $i.e.$, constructing an rGBN guided Transformer as a new larger-context neural language model.

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

## A The GBN-RNN

**GBN-RNN:** $\{y_{1:T}, \boldsymbol{d}\}$ denotes a sentence-context pair, where $\boldsymbol{d} \in \mathbb{Z}_+^{V_c}$ represents the document-level context as a word frequency count vector, the $v$th element of which counts the number of times the $v$th word in the vocabulary appears in the document excluding sentence $y_{1:T}$. The hierarchical model of a $L$-hidden-layer GBN, from top to bottom, is expressed as

$$
\boldsymbol{\theta}^L \sim \text{Gam}\left(\boldsymbol{r}, c^{L+1}\right), \ldots, \boldsymbol{\theta}^l \sim \text{Gam}\left(\boldsymbol{\Phi}^{l+1}\boldsymbol{\theta}^{l+1}, c^{l+1}\right), \ldots,
$$
$$
\boldsymbol{\theta}^1 \sim \text{Gam}\left(\boldsymbol{\Phi}^2\boldsymbol{\theta}^2, c^2\right), \ \boldsymbol{d} \sim \text{Pois}\left(\boldsymbol{\Phi}^1\boldsymbol{\theta}^1\right), \tag{9}
$$

The stacked-RNN based language model described in (3) is also used in GBN-RNN.

**Statistical inference:** To infer GBN-RNN, we consider a hybrid of stochastic gradient MCMC (Welling & Teh, 2011; Patterson & Teh, 2013; Li et al., 2015; Ma et al., 2015; Cong et al., 2017a), used for the GBN topics $\phi_k^l$, and auto-encoding variational inference (Kingma & Welling, 2013; Rezende et al., 2014), used for the parameters of both the inference network (encoder) and RNN. More specifically, GBN-RNN generalizes Weibull hybrid auto-encoding inference (WHAI) of Zhang et al. (2018): it uses a deterministic-downward-stochastic-upward inference network to encode the bag-of-words representation of $\boldsymbol{d}$ into the latent topic-weight variables $\boldsymbol{\theta}^l$ across all hidden layers, which are fed into not only GBN to reconstruct $\boldsymbol{d}$, but also a stacked RNN in language model, as shown in (3), to predict the word sequence in $y_{1:T}$. The topics $\phi_k^l$ can be sampled with topic-layer-adaptive stochastic gradient Riemannian (TLASGR) MCMC, whose details can be found in Cong et al. (2017a); Zhang et al. (2018), omitted here for brevity. Given the sampled topics $\phi_k^l$, the joint marginal likelihood of $\{y_{1:T}, \boldsymbol{d}\}$ is defined as

$$
p\left(y_{1:T}, \boldsymbol{d} \,|\, \{\boldsymbol{\Phi}^l\}_l\right) = \int p\left(\boldsymbol{d} \,|\, \boldsymbol{\Phi}^1\boldsymbol{\theta}^1\right) \left[\prod_{t=1}^T p\left(y_t \,|\, y_{1:t-1}, \boldsymbol{\theta}^{1:L}\right)\right] \left[\prod_{l=1}^L p\left(\boldsymbol{\theta}^l \,|\, \boldsymbol{\Phi}^{l+1}\boldsymbol{\theta}^{l+1}\right)\right] d_{\boldsymbol{\theta}^{1:L}}. \tag{10}
$$

For efficient inference, an inference network as $Q = \prod_{l=1}^L q(\boldsymbol{\theta}^l \,|\, \boldsymbol{d}, \boldsymbol{\Phi}^{l+1}\boldsymbol{\theta}^{l+1})$ is used to provide a ELBO of the log joint marginal likelihood as

$$
L(y_{1:T}, \boldsymbol{d}) = \mathbb{E}_Q\left[\ln p\left(\boldsymbol{d} \,|\, \boldsymbol{\Phi}^1\boldsymbol{\theta}^1\right) + \sum_{t=1}^T \ln p\left(y_t \,|\, y_{1:t-1}, \boldsymbol{\theta}^{1:L}\right)\right] - \sum_{l=1}^L \mathbb{E}_Q\left[\ln \frac{q(\boldsymbol{\theta}^l \,|\, \boldsymbol{d}, \boldsymbol{\Phi}^{l+1}\boldsymbol{\theta}^{l+1})}{p(\boldsymbol{\theta}^l \,|\, \boldsymbol{\Phi}^{l+1}\boldsymbol{\theta}^{l+1})}\right] \tag{11}
$$

and the training is performed by maximizing $\mathbb{E}_{p_{\text{data}}(y_{1:T}, \boldsymbol{d})}[L(y_{1:T}, \boldsymbol{d})]$; following Zhang et al. (2018), we define $q(\boldsymbol{\theta}^l \,|\, \boldsymbol{d}, \boldsymbol{\Phi}^{l+1}, \boldsymbol{\theta}^{l+1}) = \text{Weibull}(\boldsymbol{k}^l + \boldsymbol{\Phi}^{l+1}\boldsymbol{\theta}^{l+1}, \boldsymbol{\lambda}^l)$, where both $\boldsymbol{k}^l$ and $\boldsymbol{\lambda}^l$ are deterministically transformed from $\boldsymbol{d}$ using neural networks. Distinct from a usual variational auto-encoder whose inference network has a pure bottom-up structure, the inference network here has a determistic-upward–stoachstic-downward ladder structure (Zhang et al., 2018).

## B The coupling vector

Following Lau et al. (2017), the $\boldsymbol{a}_{j,t}^l = g^l\left(\boldsymbol{h}_{j,t}^l, \boldsymbol{\theta}_j^l\right)$ can be implemented with a gating unit similar to a GRU (Cho et al., 2014), describe as

$$
\begin{aligned}
\boldsymbol{z}_{j,t}^l &= \sigma\left(\mathbf{W}_z^l\boldsymbol{\theta}_j^l + \mathbf{U}_z^l\boldsymbol{h}_{j,t}^l + \mathbf{b}_z^l\right) \\
\boldsymbol{r}_{j,t}^l &= \sigma\left(\mathbf{W}_r^l\boldsymbol{\theta}_j^l + \mathbf{U}_r^l\boldsymbol{h}_{j,t}^l + \mathbf{b}_r^l\right) \\
\hat{\boldsymbol{h}}_{j,t}^l &= \tanh\left(\mathbf{W}_h^l\boldsymbol{\theta}_j^l + \mathbf{U}_h^l\left(\boldsymbol{r}_{j,t}^l \odot \boldsymbol{h}_{j,t}^l\right) + \mathbf{b}_h^l\right) \\
\boldsymbol{a}_{j,t}^l &= \left(1 - \boldsymbol{z}_{j,t}^l\right) \odot \boldsymbol{h}_{j,t}^l + \boldsymbol{z}_{j,t}^l \odot \hat{\boldsymbol{h}}_{j,t}^l
\end{aligned} \tag{12}
$$

## C SGMCMC for rGBN-RNN

To allow for scalable inference, we apply the topic-layer-adaptive stochastic gradient Riemannian (TLASGR) MCMC algorithm described in Cong et al. (2017a); Zhang et al. (2018), which can be used to sample simplex-constrained global parameters Cong et al. (2017b) in a mini-batch based manner. It improves its sampling efficiency via the use of the Fisher information matrix (FIM)

Girolami & Calderhead (2011), with adaptive step-sizes for the latent factors and transition matrices of different layers. In this section, we discuss how to update the global parameters $\{\boldsymbol{\Phi}^l, \boldsymbol{\Pi}^l\}_{l=1}^L$ of rGBN in detail and give a complete one in Algorithm in 1.

**Sample the auxiliary counts:** This step is about the "backward" and "upward" pass. Let us denote $Z_{\cdot kj}^l = \sum_{k_l=1}^{K_l} Z_{k_l kj}^l$, $Z_{\cdot k, J+1}^l = 0$, and $x_{kj}^{(1,1)} = d_{vj}$, where $\boldsymbol{d}_j = \{d_{1j}, .., d_{vj}, .., d_{V_cj}\}$ is shown in (2). Working backward for $j = J, ..., 1$ and upward for $l = 1, ..., L$, we draw

$$(A_{k1j}^l, ..., A_{kK_lj}^l) \sim \text{Multi}\left(x_{kj}^{(l,l)}; \frac{\phi_{k1}^l \theta_{1j}^l}{\sum_{k_l=1}^{K_l} \phi_{kk_l}^l \theta_{k_lj}^l}, ..., \frac{\phi_{kK_l}^l \theta_{K_lj}^l}{\sum_{k_l=1}^{K_l} \phi_{kk_l}^l \theta_{k_lj}^l}\right), \tag{13}$$

$$x_{kj}^{l+1} \sim \text{CRT}\left[A_{\cdot kj}^l + Z_{\cdot k, j+1}^l, \tau_0\left(\sum_{k_{l+1}=1}^{K_{l+1}} \phi_{kk_{l+1}}^{l+1} \theta_{k_{l+1}j}^{l+1} + \sum_{k_l=1}^{K_l} \pi_{kk_l}^l \theta_{k_1, j-1}^l\right)\right]. \tag{14}$$

Note that via the deep structure, the latent counts $x_{kj}^{l+1}$ will be influenced by the effects from both time $j-1$ at layer $l$ and time $j$ at layer $l+1$. With $p_1 := \sum_{k_l=1}^{K_l} \pi_{kk_l}^l \theta_{k_lj-1}^l$ and $p_2 := \sum_{k_{l+1}=1}^{K_{l+1}} \phi_{kk_{l+1}}^{l+1} \theta_{k_{l+1}j}^{l+1}$, we can sample the latent counts at layer $l$ and $l+1$ by

$$(x_{kj}^{l+1,l}, x_{kj}^{l+1,l+1}) \sim \text{Multi}\left(x_{kj}^{l+1}, p_1/(p_1+p_2), p_2/(p_1+p_2)\right), \tag{15}$$

and then draw

$$(Z_{k1j}^l, ..., Z_{kK_lj}^l) \sim \text{Multi}\left(x_{kj}^{l+1,l}; \frac{\pi_{k1}^l \theta_{1,j-1}^l}{\sum_{k_l=1}^{K_l} \pi_{kk_l}^l \theta_{k_l,j-1}^l}, ..., \frac{\pi_{kK_l}^l \theta_{K_l,j-1}^l}{\sum_{k_l=1}^{K_l} \pi_{kk_l}^l \theta_{k_l,j-1}^{(l)}}\right). \tag{16}$$

In rGBN, the prior and the likelihood of $\{\boldsymbol{\Phi}^l\}_{l=1}^L$ is very similar with $\{\boldsymbol{\Pi}^l\}_{l=1}^L$, so we also apply the TLASGR MCMC sampling algorithm on both of them conditioned on the auxiliary counts.

**Sample the hierarchical components $\{\boldsymbol{\Phi}^l\}_{l=1}^L$:** For $\boldsymbol{\phi}_k^l$, the $k$th column of the loading matrix $\boldsymbol{\Phi}^l$ of layer $l$, its sampling can be efficiently realized as

$$(\boldsymbol{\phi}_k^l)_{n+1} = \left[(\boldsymbol{\phi}_k^l)_n + \frac{\varepsilon_n}{P_k^l}\left[\left(\rho\tilde{\boldsymbol{A}}_{\cdot k\cdot}^l + \eta_0^l\right) - \left(\rho\tilde{A}_{\cdot k\cdot}^l + K_{l-1}\eta_0^l\right)(\boldsymbol{\phi}_k^l)_n\right]\right.$$
$$\left. + \mathcal{N}\left(0, \frac{2\varepsilon_n}{P_k^l}\left[\text{diag}(\boldsymbol{\phi}_k^l)_n - (\boldsymbol{\phi}_k^l)_n(\boldsymbol{\phi}_k^l)_n^T\right]\right)\right]_{\angle}, \tag{17}$$

where $P_k^l$ is calculated using the estimated FIM, $\tilde{A}_{k_lj\cdot}^l = \sum_{j=1}^J A_{k_lkj}^l$, $\tilde{\boldsymbol{A}}_{\cdot k\cdot}^l = \{\tilde{A}_{1j\cdot}^l, \cdots, \tilde{A}_{K_lj\cdot}^l\}^T$ and $\tilde{A}_{\cdot k\cdot}^l = \sum_{k_l=1}^{k_l} \tilde{A}_{k_lj\cdot}^l$, $A_{k_lkj}^l$ comes from the augmented latent counts $A^l$ in (13), $\eta_0^l$ denote the prior of $\boldsymbol{\phi}_k^l$, and $[\cdot]_{\angle}$ denotes a simplex constraint.

**Sample the transmission matrix $\{\boldsymbol{\Pi}^l\}_{l=1}^L$:** For $\boldsymbol{\pi}_k^l$, the $k$th column of the transition matrix $\boldsymbol{\Pi}^l$ of layer $l$, its sampling can be efficiently realized as

$$(\boldsymbol{\pi}_k^l)_{n+1} = \left[(\boldsymbol{\pi}_k^l)_n + \frac{\varepsilon_n}{M_k^l}\left[\left(\rho\tilde{\boldsymbol{Z}}_{\cdot k\cdot}^l + \boldsymbol{\eta}_{\cdot k}^l\right) - \left(\rho\tilde{Z}_{\cdot k\cdot}^l + \eta_{\cdot k}^l\right)(\boldsymbol{\pi}_k^l)_n\right]\right.$$
$$\left. + \mathcal{N}\left(0, \frac{2\varepsilon_n}{M_k^l}\left[\text{diag}(\boldsymbol{\pi}_k^l)_n - (\boldsymbol{\pi}_k^l)_n(\boldsymbol{\pi}_k^l)_n^T\right]\right)\right]_{\angle}, \tag{18}$$

where $M_k^l$ is calculated using the estimated FIM, $\tilde{Z}_{k_lj\cdot}^l = \sum_{j=1}^J Z_{k_lkj}^l$, $\tilde{\boldsymbol{Z}}_{\cdot k\cdot}^l = \{\tilde{Z}_{1j\cdot}^l, \cdots, \tilde{Z}_{K_lj\cdot}^l\}^T$ and $\tilde{Z}_{\cdot k\cdot}^l = \sum_{k_l=1}^{k_l} \tilde{Z}_{k_lj\cdot}^l$, $Z_{k_lkj}^l$ comes from the augmented latent counts $Z^l$ in (16), and $[\cdot]_{\angle}$ denotes a simplex constraint, and $\boldsymbol{\eta}_{\cdot k}^l$ denotes the prior of $\boldsymbol{\pi}_k^l$, more details about TLASGR-MCMC for our proposed model can be found in Cong et al. (2017a).

# D  DATASETS

We consider three publicly available corpora[1]. APNEWS is a collection of Associated Press news articles from 2009 to 2016, IMDB is a set of movie reviews collected by Maas et al. (2011), and BNC

---

[1]https://ibm.ent.box.com/s/ls61p8ovc1y87w45oa02zink2zl7l6z4

is the written portion of the British National Corpus (Consortium, 2007). Following the preprocessing steps in Lau et al. (2017), we tokenize words and sentences using Stanford CoreNLP (Klein & Manning, 2003), lowercase all word tokens, and filter out word tokens that occur less than 10 times. For the topic model, we additionally exclude stopwords[2] and the top $0.1\%$ most frequent words. All these corpora are partitioned into training, validation, and testing sets, whose summary statistics are provided in Table 2 of the Appendix.

Table 2: Summary statistics for the datasets.

| Dataset | Vocubalry | | Training | | | Validation | | | Testing | | |
|---|---|---|---|---|---|---|---|---|---|---|---|
| | LM | TM | Docs | Sents | Tokens | Docs | Sents | Tokens | Docs | Sents | Tokens |
| APNEWS | 34231 | 32169 | 50K | 0.8M | 15M | 2K | 33K | 0.6M | 2K | 32K | 0.6M |
| IMDB | 36009 | 34925 | 75K | 1.1M | 20M | 12.5K | 0.18M | 0.3M | 12.5K | 0.18M | 0.3M |
| BNC | 43703 | 41552 | 15K | 1M | 18M | 1K | 57K | 1M | 1K | 66K | 1M |

# E    COMPLEXITY OF RGBN-RNN

The proposed rGBN-RNN consists of both language model and topic model components. For the topic model component, there are the global parameters of rGBN (decoder), including $\{\Phi^l, \Pi^l\}_{l=1}^L$ in (2) , and the parameters of the variational recurrent inference network (encoder), consisting of $\text{RNN}_{\text{sent}}^l$, $f_{\boldsymbol{k}}^l$, and $f_{\boldsymbol{\lambda}}^l$ in (8). The language model component is parameterized by $\text{LSTM}_{\text{word}}^l$ in (3) and the coupling vectors $g^l$ described in Appendix B. We summarize in Table 3 the complexity of rGBN-RNN (ignoring all bias terms), where $V$ denotes the vocabulary size of the language model, $E$ the dimension of word embedding vectors, $V_c$ the size of the vocabulary of the topic model that excludes stop words, $H_l^w$ the number of hidden units of the word-level LSTM at layer $l$ (stacked-RNN language model), $H_l^s$ the number of hidden units of the sentence-level RNN at layer $l$ (variational recurrent inference network), and $K_l$ the number of topics at layer $l$.

Table 4 further compares the number of parameters between various RNN-based language models, where we follow the convention to ignore the word embedding layers. Some models in Table 1 are not included here, because we could not find sufficient information from their corresponding papers or code to accurately calculate the number of model parameters. Note when used for language generation at the testing stage, rGBN-RNN no longer needs its topics $\{\Phi^l\}$, whose parameters are hence not counted. Note the number of parameters of the topic model component is often dominated by that of the language model component.

Table 3: Complexity of the three-layer rGBN-RNN.

| Component | Language Model | | Topic Model | | | | |
|---|---|---|---|---|---|---|---|
| Param | $\text{LSTM}_{\text{word}}^l$ in (3) | $g^l$ in B | $\Phi^l$ in (2) | $\Pi^l$ in (2) | $\text{RNN}_{\text{sent}}^l$ in (8) | $f_{\boldsymbol{k}}^l$ in (8) | $f_{\boldsymbol{\lambda}}^l$ in (8) |
| Layer1 | $O(4 \times (E + H_1^w) \times H_1^w)$ | $O(3 \times (K_1 + H_1^w) \times H_1^w)$ | $O(V_c \times K_1)$ | $O(K_1 \times K_1)$ | $O((V_c + H_1^s) \times H_1^s)$ | $O(H_1^s)$ | $O(K_1 \times H_1^s)$ |
| Layer2 | $O(4 \times (H_1^w + H_2^w) \times H_2^w)$ | $O(3 \times (K_2 + H_2^w) \times H_2^w)$ | $O(K_1 \times K_2)$ | $O(K_2 \times K_2)$ | $O((H_1^s + H_2^s) \times H_2^s)$ | $O(H_2^s)$ | $O(K_2 \times H_2^s)$ |
| Layer3 | $O(4 \times (H_2^w + H_3^w) \times H_3^w)$ | $O(3 \times (K_3 + H_3^w) \times H_3^w)$ | $O(K_2 \times K_3)$ | $O(K_3 \times K_3)$ | $O((H_2^s + H_3^s) \times H_3^s)$ | $O(H_3^s)$ | $O(K_3 \times H_3^s)$ |

Table 4: Comparison of the number of parameters of different models when used for language generation.

| Model | LSTM Size | Topic Size | # LM Param | #TM Param | # All Param |
|---|---|---|---|---|---|
| TDLM (Lau et al., 2017) | 600 | 100 | 3.35M | 0.019M | 3.37M |
| | 900-900 | 100 | 13.38M | 0.019M | 13.40M |
| basic-LSTM (Hochreiter & Schmidhuber, 1997) | 600 | — | 2.16M | — | 2.16M |
| | 900-900 | — | 10.80M | — | 10.80M |
| | 900-900-900 | — | 17.68M | — | 17.68M |
| GBN-RNN | 600 | 100 | 3.40M | 0.02M | 3.42M |
| | 600-512 | 100-80 | 6.50M | 0.04M | 6.54M |
| | 600-512-256 | 100-80-50 | 7.20M | 0.05M | 7.25M |
| rGBN-RNN | 600 | 100 | 3.40M | 0.03M | 3.43M |
| | 600-512 | 100-80 | 6.50M | 0.06M | 6.56M |
| | 600-512-256 | 100-80-50 | 7.20M | 0.07M | 7.27M |

---

[2]We use Mallet's stopword list: https://github.com/mimno/Mallet/tree/master/stoplists.

## F   More experimental results on IMDB and BNC

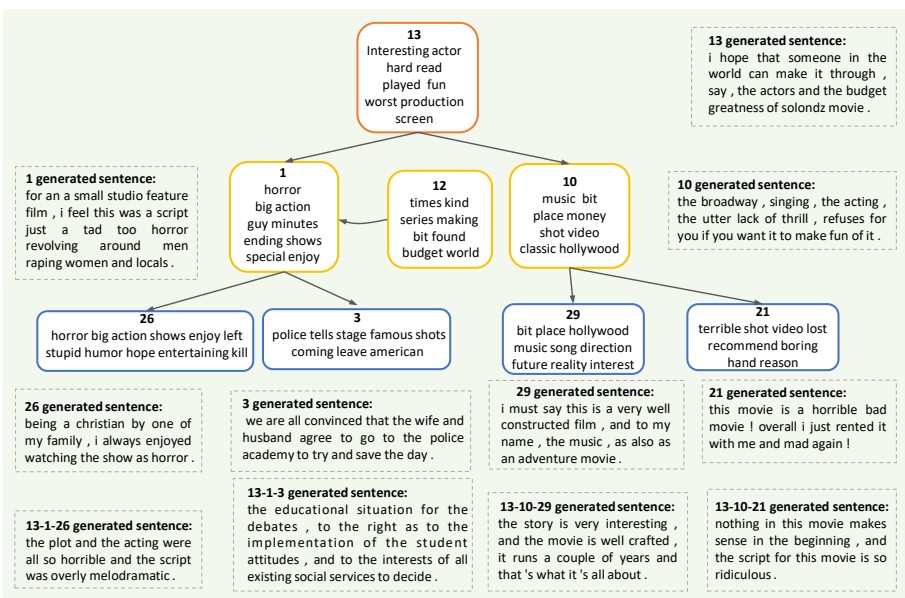

Figure 6: Topics and their temporal trajectories inferred by a three-hidden-layer rGBN-RNN from the IMDB dataset, and the generated sentences under topic guidance (best viewed in color). Top words of each topic are shown in orange, yellow and blue box, and each sentence is shown in a dotted line box labeled with the corresponding topic index. Sentences generated with a combination of topics in different layers are at the bottom of the figure.

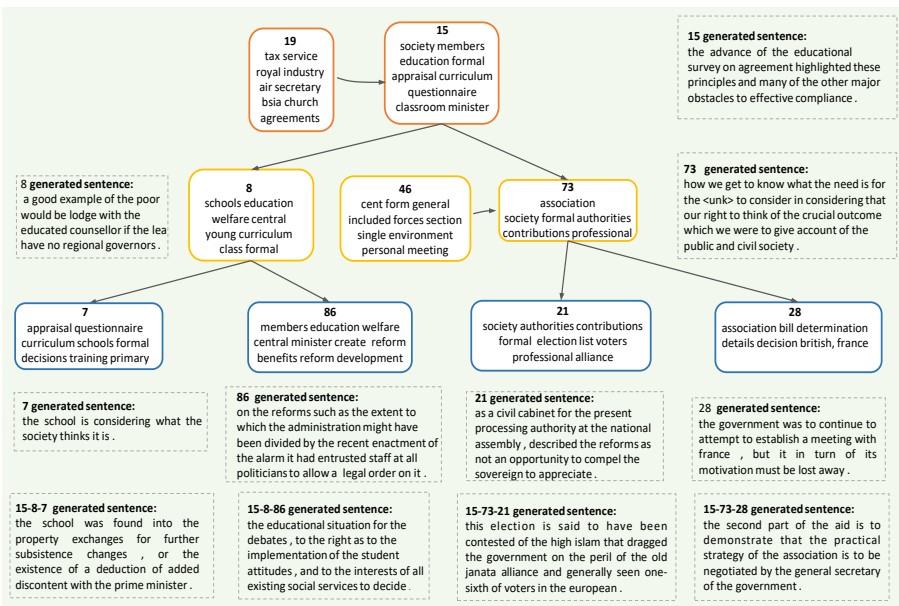

Figure 7: Topics and their temporal trajectories inferred by a three-hidden-layer rGBN-RNN from the BNC dataset, and the generated sentences under topic guidance (best viewed in color). Top words of each topic are shown in orange, yellow and blue box, and each sentence is shown in a dotted line box labeled with the corresponding topic index. Sentences generated with a combination of topics in different layers are at the bottom of the figure.

# G    BLEU SCORES FOR IMDB

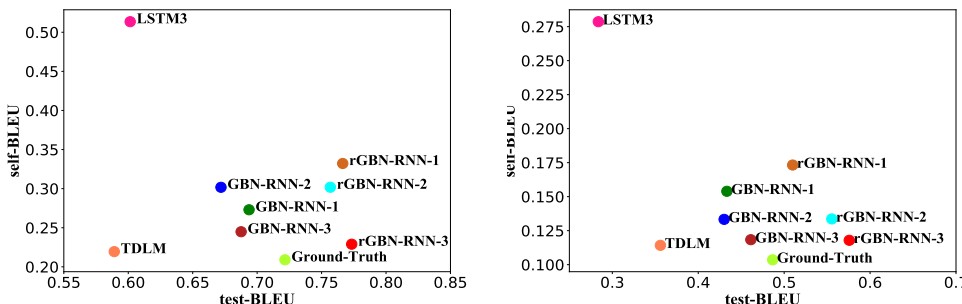

Figure 8: BLEU scores of different methods for IMDB. x-axis denotes test-BLEU, and y-axis self-BLEU. Left panel is BLEU-3 and right is BLEU-4, and a better BLEU score would fall within the lower right corner, where black point represents mean value and circles with different colors denote the elliptical surface of probability of BLEU in a two-dimensional space.

# H    BLEU SCORES FOR APNEWS

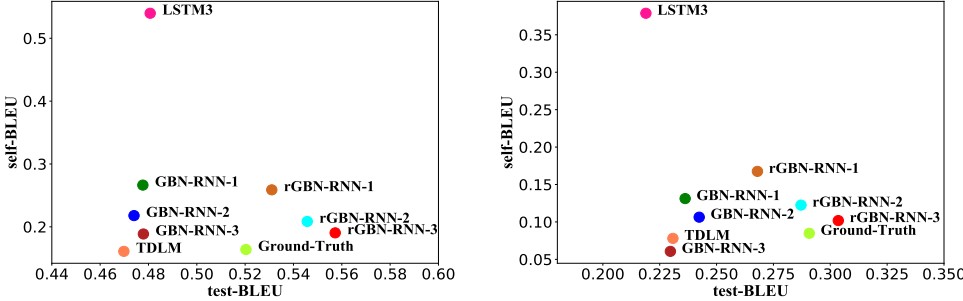

Figure 9: BLEU scores of different methods for APNEWS. x-axis denotes test-BLEU, and y-axis self-BLEU. Left panel is BLEU-3 and right is BLEU-4, and a better BLEU score would fall within the lower right corner, where black point represents mean value and circles with different colors denote the elliptical surface of probability of BLEU in a two-dimensional space.

