# OpenReview forum: "Recurrent Hierarchical Topic-Guided Neural Language Models"
_ICLR.cc/2020/Conference — Reject_

### Official Review · AnonReviewer2 · 2019-10-24
**Official Blind Review #2**

**Rating:** 8

**Review:**

This paper presents rGBN-RNN, a model that integrates a hierarchical recurrent topic model with an RNN-based language model in order to incorporate global semantic information and improve capturing of inter-sentence relations. The proposed model improves in perplexity across the three tested datasets over state of the art models of comparable type, and follow-up analyses show strong performance in sentence and paragraph generation, as well as learning of sensible hierarchical topics.

Overall I think this is a clearly-written paper with a well-motivated and interesting model, strong results, and a good range of follow-up analyses. I think that it is a solid paper to accept for publication.

Some areas for improvement:

It seems strange not to mention all of the recent high-profile work on LM-based pre-training, since my impression is that these models operate effectively with large multi-sentence contexts. Do models like BERT and GPT-2 fail to take into account inter-sentence relations, as the paper claims most LMs do? I would like to see more discussion of how this work fits with that.

I don't know that it makes sense to highlight as the contribution of this model that it can "simultaneously capture syntax and semantics". It's not clear to me that other language models fail to capture semantics (keeping in mind that semantics applies within a sentence and not just at a global level) -- rather, it seems that the strength of this model is in capturing semantic relations above the sentence level.  If this is correct, that should be expressed more precisely.

It's not clear to me what we learn from Figure 3. The claim is that "the color of the hidden states of the stacked RNN based language model at layer 1 changes quickly ... because lower layers are in charge of learning short-term dependencies", but looking at the higher layers I'm not seeing clear evidence of capturing of long-distance dependencies, or even clear capturing of syntactic constituents. The takeaways from that figure should be made clearer and should be sure to correspond to what we can actually confidently conclude from that analysis.


**Experience Assessment:**

I have read many papers in this area.

**Review Assessment: Checking Correctness Of Derivations And Theory:**

I did not assess the derivations or theory.

**Review Assessment: Checking Correctness Of Experiments:**

I assessed the sensibility of the experiments.

**Review Assessment: Thoroughness In Paper Reading:**

I made a quick assessment of this paper.

---

> ### Author Response · Authors · 2019-11-15
> **We have made the suggested improvements**
>
> Thank you for your comments and suggestions. We have revised our paper accordingly and highlighted the main changes in red.
>
> Q1: It seems strange not to mention all of the recent high-profile work on LM-based pre-training, since my impression is that these models operate effectively with large multi-sentence contexts. Do models like BERT and GPT-2 fail to take into account inter-sentence relations, as the paper claims most LMs do? I would like to see more discussion of how this work fits with that.
>
> A1: Thanks for your suggestion and we have now added related discussion about the recent high-profile work on LM-based pre-training (e.g., in the third paragraph of Section 3.1). First, while Transformer based LMs have been shown to be powerful, they often have significantly more parameters and require significantly more computation to train than RNN based LMs. Second, we do not view rGBN-RNN and Transformer based LMs as directly comparable given their differences in sizes and structures; after carefully comparing rGBN-RNN with Transformer based LMs, we believe a promising future extension of rGBN-RNN is to replace its stacked RNN with a Transformer-based LM, i.e., constructing a rGBN guided Transformer (rGBN-Transformer). Note the AC had made related comments, to which we had provided detailed response; please see that response for more details.
>
> Q2: I don’t know that it makes sense to highlight as the contribution of this model that it can "simultaneously capture syntax and semantics". It’s not clear to me that other language models fail to capture semantics (keeping in mind that semantics applies within a sentence and not just at a global level) – rather, it seems that the strength of this model is in capturing semantic relations above the sentence level. If this is correct, that should be expressed more precisely.
>
> A2: Thank you for your insightful comments. We have changed “semantics” to “global semantics” to more precisely express the idea that rGBN-RNN can capture semantic relations above the sentence level.
>
> Q3:  It’s not clear to me what we learn from Figure 3. The claim is that "the color of the hidden states of the stacked RNN based language model at layer 1 changes quickly ... because lower layers are in charge of learning short-term dependencies", but looking at the higher layers I’m not seeing clear evidence of capturing of long-distance dependencies, or even clear capturing of syntactic constituents. The takeaways from that figure should be made clearer and should be sure to correspond to what we can actually confidently conclude from that analysis.
>
> A3:  Thank you for your suggestion. We have carefully revised the first paragraph of Section 3.2 to explain in detail how to interpret Figure 3 (by comparing the segments exhibited in the L2-norm sequence of a hidden layer to the corresponding words in the sentence), which now more clearly supports our claim.

---

### Official Review · AnonReviewer3 · 2019-10-25
**Official Blind Review #3**

**Rating:** 8

**Review:**


The paper  proposes deep recurrent topic model guided language modeling using a stacked RNN, and uses a novel variational recurrent inference network to learn the parameters.  The proposed model can capture the dependence across the sentences in language generation though the recurrent latent topics. Moreover, the deep rGBN architecture provides Gamma distributed topic topic weight vectors which can be associated with every layer of the stacked RNN generating the sentence. The parameters of both the hierarchical recurrent topic model and language model are learnt using a hybrid inference algorithm  combining  variational inference to estimate language model and inference network parameters and MCMC to infer rGBN parameters. The effectiveness of the proposed model on the language modeling task is demonstrated on 3 datasets using Perplexity and BLEU score. The paper also provides a visual representation of the topics and their temporal trajectories.

The proposed model extends previous approaches on topic guided language modeling by using deep rGBN model. Though the novelty of the model is limited, learning and inference with the proposed model is non-trivial. Further, the paper show an improvement in performance on language modeling using the proposed approach over SOTA approaches, demonstrating the significance of the proposed approach.

Though the paper is relatively well written, it would have been good to explain some points on architecture and inference. It would have been better to provide the rationale behind some architectural decisions like associating \theta^1 and g^1 as against g^3. Related to this, Figure 1 has a typo where \theta^2 is associated with g^3.  An explanation on combining all the  latent representation in the RNN model used for language modeling will be helpful, though this is motivated by previous approaches. A proper explanation the TLASGR-MCMC approach for sampling from the posterior of  rGBN parameters is missing in the main paper.  It would be good to provide some details of this in the main paper.

Experimental section compares the proposed approach against many SOTA approaches for the language modeling task. It would have been good to provide a quantitive evaluation of the topic modeling task also  in addition to demonstrating them qualitatively.

**Experience Assessment:**

I have read many papers in this area.

**Review Assessment: Checking Correctness Of Derivations And Theory:**

I assessed the sensibility of the derivations and theory.

**Review Assessment: Checking Correctness Of Experiments:**

I assessed the sensibility of the experiments.

**Review Assessment: Thoroughness In Paper Reading:**

I read the paper at least twice and used my best judgement in assessing the paper.

---

> ### Author Response · Authors · 2019-11-15
> **More explanations and details are now included**
>
> Thank you for your comments and suggestions. We have revised our paper accordingly and highlighted the main changes in red.
>
> Q1: Though the paper is relatively well written, it would have been good to explain some points on architecture and inference. It would have been better to provide the rationale behind some architectural decisions like associating \theta^1 and g^1 as against g^3. Related to this, Figure 1 has a typo where \theta^2 is associated with g^3.
>
> A1: Thank you for your insightful comments, which make us realize that the upward arrows in Fig. 1(b) and/or the upward red arrows in Fig. 1(c) can be flipped, leading to three additional architectural variations of the proposed rGBN-RNN. Since the current rGBN-RNN already clearly outperforms other RNN based models, given both the space and time constraints, we leave these three architectural variations of rGBN-RNN to future study. Also thank you for catching the typo! We have fixed it.
>
> Q2: An explanation on combining all the latent representation in the RNN model used for language modeling will be helpful, though this is motivated by previous approaches.
>
> A2: We have added two reasons to explain why combining all the latent representation to help language modeling. Please see the paragraph below Equation (4) for detail.
>
> Q3: A proper explanation the TLASGR-MCMC approach for sampling from the posterior of rGBN parameters is missing in the main paper. It would be good to provide some details of this in the main paper.
>
> A3: We have added more details on TLASGR-MCMC to Appendix C. For now, we have kept all the details of TLASGR-MCMC in Appendix C due to the space constraint (we find it difficult to move only one or two equations back to the main paper while maintaining the clarity of the technical details). We are open to your further suggestions on this.
>
> Q4: Experimental section compares the proposed approach against many SOTA approaches for the language modeling task. It would have been good to provide a quantitive evaluation of the topic modeling task also in addition to demonstrating them qualitatively.
>
> A4: We have not provided quantitative evaluation of the topic modeling task mainly for two reasons. First, as our paper is focused on improving an RNN-based language model with a deep dynamic topic model, adding that evaluation may distract it from the main purpose. Second, our topics learned by rGBN-RNN are hierarchical between different layers and recurrent at the same layer. Thus it is unclear whether existing quantitative measures (e.g., topic coherence), which are often designed to evaluate conventional single-layer topic model without temporal structure, would be appropriate to evaluate the quality of the rGBN topics that are both hierarchical and temporally linked.

---

### Official Review · AnonReviewer1 · 2019-10-28
**Official Blind Review #1**

**Rating:** 8

**Review:**

This paper presents a method for natural language generation, using a language model, informed by a topic model.
The topic model is a hierarchical recurrent topic model that attempts to extract document-level word concurrence patterns and topic weight vectors for sentences.
The language model is a stacked RNN model, aiming to capture word sequential dependencies.

The proposed method is a combination of two existing methods, i.e. gamma-belief networks  and stacked RNN, where the stacked RNN is improved with the information from recurrent gamma belief network.

Overall, this is a well written paper, clearly presented, with certain novelties. The method is well formulated mathematically and evaluated experimentally. The results look interesting especially for capturing the long-range dependencies, as shown by the  BLEU scores. One suggestion is that the authors didn't include computational analysis about the complexity and loads of the proposed method as compared with the baseline methods.

**Experience Assessment:**

I have read many papers in this area.

**Review Assessment: Checking Correctness Of Derivations And Theory:**

I carefully checked the derivations and theory.

**Review Assessment: Checking Correctness Of Experiments:**

I assessed the sensibility of the experiments.

**Review Assessment: Thoroughness In Paper Reading:**

I read the paper thoroughly.

---

> ### Author Response · Authors · 2019-11-15
> **Complexity analysis has been added to the Appendix**
>
> Thank you for your positive feedback. We have revised our paper accordingly and highlighted the main changes in red.
>
> Q: One suggestion is that the authors didn’t include computational analysis about the complexity and loads of the proposed method as compared with the baseline methods.
>
> A: Thank you for your suggestion. We have revised our paper to include a comprehensive complexity analysis, as shown in Appendix E. Examining Table 1 and the newly added Table 4 in Appendix E, one can find that the proposed rGBN-RNN achieves better performance with fewer parameters than comparable baselines. We note that when rGBN-RNN is used as a language model after training (which means the inferred topics $\Phi$ are no longer needed), the number of parameters of the rGBN topic model component is dominated by that of the RNN language model component.

---

### Official Review · AnonReviewer5 · 2019-12-02
**Official Blind Review #5**

**Rating:** 1

**Review:**

The model description is confusing and lots of statements are presented without appropriate or enough justification. For example, (1) in the last paragraph of page 2, they claimed that the language component is used in their model to capture syntactic information, which I do not feel comfortable to accept; (2) in the first paragraph of page 3, it says "we define d_j as the BoW vector summarizing only the preceding sentences", without further information, I have no idea what a BoW vector looks like or how it is constructed; (3) in the last paragraph of page 3, it says using Dirichlet priors to make "the latent representation more identifiable and interpretable, but also facilitates inference", which I really don't know what it means. There are a few more examples like these.

More importantly, I think Eq. (5) is wrong, which makes me question their whole methodology. To be specific, in their definition, d_j refers to a summary of all the sentences other than s_j. That means,
- for s_1, d_1 is defined on s_2, s_3, s_4, ..., s_J; and
- for s_2, d_2 is defined on s_1, s_3, s_4, ..., s_J.
In other words, there is a huge overlap between any two d_j and d_{j'}. Therefore, I am not sure the decomposition on the right hand side of equation 5 (particularly, the decomposition of p(d_j | ...) ) is valid.

Although they have some interesting results and the lowest PPLx comparing to other models, I do not think this paper is ready to be accepted.

**Experience Assessment:**

I have published in this field for several years.

**Review Assessment: Checking Correctness Of Derivations And Theory:**

I carefully checked the derivations and theory.

**Review Assessment: Checking Correctness Of Experiments:**

I carefully checked the experiments.

**Review Assessment: Thoroughness In Paper Reading:**

I read the paper thoroughly.

---

> ### Author Response · Authors · 2019-12-20
> **Response to Review #5**
>
> We strongly argue against the comments of Reviewer 5 (R5).
>
> (1)  To make R5 feel more comfortable about our claim that the RNN-based language model component is used to capture syntactic information, we'd like to point out similar claims in the literature: TopicRNN [1] claims in its abstract that "TopicRNN model integrates the merits of RNNs and latent topic models: it captures local (syntactic) dependencies using an RNN and global (semantic) dependencies using latent topics;” TCNLM [2] claims in its conclusion that "the topic model part captures the global semantic meaning in a document, while the language model part learns the local semantic and syntactic relationships between words." We could have revised this claim if R5 could explain why he/she felt uncomfortable about it.
>
> [1] Adji B Dieng, Chong Wang, Jianfeng Gao, and John Paisley. TopicRNN: A recurrent neural network with long-range semantic dependency. In ICLR, 2017.
>
> [2] Wenlin Wang, Zhe Gan, Wenqi Wang, Dinghan Shen, Jiaji Huang, Wei Ping, Sanjeev Satheesh, and Lawrence Carin. Topic compositional neural language model. In AISTATS, pp. 356–365, 2018
>
> (2) We are very surprised to read R5's comment: "I have no idea what a BoW vector looks like or how it is constructed," especially considering that R5 claimed: "I have published in this field for several years." Please allow us to explain a well-known concept in text analysis and retrieval: denoting V as the vocabulary size, a bag-of-words (BoW) vector is a V-dimensional term-frequency count vector, whose $v$th element counts the number of times the $v$th term in the vocabulary appears in a document (or a set sentences); while it completely ignores the word order, it is well suited to capture document-level word concurrence patterns (topics).
>
> (3) A Dirichlet prior imposes a simplex constraint (nonnegative elements + unit L1 norm of the vector) and often encourages sparsity (to aid both interpretability and identifiability). It also facilitates posterior inference as the conditional posterior of each column of \Phi also follows the Dirichlet distribution after performing appropriate variable augmentation (this property has been exploited by [3] to derive Gibbs sampling and [4] to derive TLASGR-MCMC, an efficient stochastic-gradient MCMC under the simplex constraint). We consider these as well-known concepts in topic modeling related literature and hence unnecessary to provide that detailed explanations.
>
> [3] Mingyuan Zhou, Yulai Cong, and Bo Chen. Augmentable gamma belief networks. J. Mach. Learn. Res., 17(163):1–44, 2016.
>
> [4] Yulai Cong, Bo Chen, Hongwei Liu, and Mingyuan Zhou. Deep latent Dirichlet allocation with topic-layer-adaptive stochastic gradient Riemannian MCMC. In Proc. of ICML 2017
>
> (4) We don't understand why you consider Eq. (5) as wrong. d_j will be the BoW vector extracted from all sentences in a document except sentence s_j. We model d_j under the Poisson factor analysis likelihood as p(d_j | ...) = Poisson(d_j; \Phi^(1) \theta_j^(1)). While Eq. 5 is not the marginal likelihood of an "exact" generative model, due to the overlap between the d_j's and the words y_jt's, it is perfectly valid to help introduce the objective function (the ELBO in Eq. 6) to be optimized in this paper.
>
> We also note at the testing stage, if the task is solely for language generation, then the topic modeling component in the decoder will be discarded, and d_j will only consist of s_1,...,s_{j-1}.
>
> We further note we submitted the code, with which you can verify the technical details and experimental results.
>
> We emphasize while we focus on guiding (stacked-)RNN with GBN or rGBN, the same idea can be adapted to potentially improve a Transformer based model with the help of GBN or rGBN. This extension, which is non-trivial at all due to the size of Transformer, is beyond the scope of this paper. With that said, motivated by the AC’s comments, we have been working on GBN-Transformer and rGBN-Transformer and we hope we can report comprehensive results now, but Transformer is so computationally demanding to train that our pace of progress has been limited by our current computational resource.

---

### Official Review · AnonReviewer4 · 2019-12-02
**Official Blind Review #4**

**Rating:** 1

**Review:**

[Additional review]
This paper proposes a technique to incorporate document-level topic model information into language models.

While the underlying idea is interesting, my biggest issue is with the misleading assertions at the very beginning of the paper. In the second paragraph of Section 1, the paper claims that RNN-based LMs often make independence assumptions between sentences, hence why they develop a topic modelling approach to model document-level information. Some issues with this claim:

1. Pretty much every LM paper that evaluates on language modelling benchmark (PTB, WT-103, Wikitext-2) uses LSTMs/Transformers incorporate cross-sentential, document-level information as context, through a very simple approach of just concatenating all the sentences and adding a unique token to mark sentence boundaries.

2. Prior work has shown that LSTMs/Transformers with cross-sentential context can, and in fact do, make use of information from previous sentences.

a. Evidence 1: Khandelwal et al. (2018) showed that LSTMs memorise word orders from the past ~50 tokens, and retain semantic information from the past ~200 tokens; both of which extend far beyond the length of an average sentence, suggesting that information from the previous sentences is used in the predictions of the current sentence.

b. Evidence 2: Language models that operate on single sentences typically do worse than language models that take into account cross-sentential context, e.g. the language model of Kim et al. (2019) that operates on single sentences gets ~90 ppl. on PTB test set, while LSTMs that condition on multiple sentences get a much better ~50-something ppl. on Mikolov PTB.

Crucially, these prior works defeat the paper’s motivation of why it claims to need topic models in the first place (i.e. to model cross-sentential context), while just concatenating multiple sentences as context would do, and in fact has been done many times.

2. Prior work (mostly in Transformer-land) has come up with ways to make use of very long-range context, from Transformer-XL to the more recent compressive Transformer (https://openreview.net/forum?id=SylKikSYDH) that can condition on entire books. While these are done for Transformers, in principle one can also apply similar techniques to LSTMs.

3. While Transformer-XL has the potential to make use of word orders in the preceding sentences, it seems that this paper’s approach cannot do that, since they only take the bag-of-words from the preceding sentences. It thus seems that their bag-of-word approach is less expressive, and hence less powerful, than the simpler alternative of concatenating sentences.

4. The perplexity results (Table 1) are not done on very standard datasets (no PTB evaluation for instance). It is thus hard to evaluate the strength of the baseline models. In the paper's defense, it seems that they were following the experimental setup of Wang et al. (2019), but the paper should elaborate more on the choice of evaluation datasets.

5. The inference part is not particularly self-contained. The paper simply refers the TLASGR-MCMC method (which is an important part to make inference scalable) to prior work (Cong et al., 2017; Zhang et al., 2018), yet does not explain (even briefly) how the approach works, and how it can be combined with their recurrent topic model formulation.

6. Evaluation of the induced topic hierarchy (Figure 4) is only done through qualitative samples, and the paper does not really explain how to pick the samples (i.e. possible cherry-picking). I am not very familiar with the topic modelling literature, but it would be nice if the induced hierarchy can be evaluated quantitatively.

References:
1. Urvashi Khandelwal, He He, Peng Qi, and Dan Jurafsky. Sharp nearby, fuzzy far away. In Proc. of ACL 2018.
2. Yoon Kim, Alexander Rush, Lei Yu, Adhiguna Kuncoro, Chris Dyer, and Gabor Melis. Unsupervised recurrent neural network grammars. In Proc. of NAACL 2019.
3. Wenlin Wang, Zhe Gan, Hongteng Xu, Ruiyi Zhang, Guoyin Wang, Dinghan Shen, Changyou Chen, and Lawrence Carin. Topic-guided variational autoencoders for text generation. In Proc. of NAACL 2019.
4. Yulai Cong, Bo Chen, Hongwei Liu, and Mingyuan Zhou. Deep latent Dirichlet allocation with topic-layer-adaptive stochastic gradient Riemannian MCMC. In Proc. of ICML 2017
5. Hao Zhang, Bo Chen, Dandan Guo, and Mingyuan Zhou. WHAI: Weibull hybrid autoencoding inference for deep topic modeling. In Proc. of ICLR 2018

**Experience Assessment:**

I have read many papers in this area.

**Review Assessment: Checking Correctness Of Derivations And Theory:**

I assessed the sensibility of the derivations and theory.

**Review Assessment: Checking Correctness Of Experiments:**

I assessed the sensibility of the experiments.

**Review Assessment: Thoroughness In Paper Reading:**

I read the paper at least twice and used my best judgement in assessing the paper.

---

> ### Author Response · Authors · 2019-12-20
> **Response to Review #4 (Part 1)**
>
> We strongly argue against the comments of Reviewer 4 (R4).
>
> 1. R4 dismissed the whole idea of using a topic modeling approach to model document-level information because “Pretty much every LM paper … uses LSTMs/Transformers incorporate cross-sentential, document-level information as context, through a very simple approach of just concatenating all the sentences and adding a unique token to mark sentence boundaries.”
>
> Response 1: Yes, this is such a simple approach, but does it work well and solve all the problems? If this simple approach works so well, why should Dieng et al. (ICLR 2018) even bother to propose topic-RNN, Wang et al. (AISTATS 2018, NAACL 2019) propose topic guided language models, and Dai et al. (ACL 2019) propose Transformer-XL? We recommend R4 to at least take a look at Related Work section of Dai et al. (ACL 2019), from which we quote one sentence: “More broadly, in generic sequence modeling, how to capture long-term dependency has been a long-standing research problem.” In other words, simply concatenating sentences has not yet solved the problem of capturing long-term dependency.
>
> 2. R4 commented that “Prior work has shown that LSTMs/Transformers with cross-sentential context can, and in fact do, make use of information from previous sentences,” provided two evidences, and claimed that “these prior works defeat the paper’s motivation of why it claims to need topic models in the first place (i.e. to model cross-sentential context), while just concatenating multiple sentences as context would do, and in fact has been done many times.”
>
> Response 2: Related to Response 1, concatenating multiple sentences as the input is indeed simple, but at what cost and how effective? One cost is the model size may quickly increase with the input sequence length, making it hungrier for computation, memory, and data size, and more difficult to train. We don’t understand why the existence of a remedy to model cross-sentential context can be used to defeat our motivation of using topic models to capture document-level semantic information, with which each input to the proposed lager-context language model can be as short as a single sentence.
>
> 2.1. Prior work (mostly in Transformer-land) has come up with ways to make use of very long-range context, from Transformer-XL to the more recent compressive Transformer (https://openreview.net/forum?id=SylKikSYDH) that can condition on entire books. While these are done for Transformers, in principle one can also apply similar techniques to LSTMs.
>
> Response 2.1: While in principle one can also apply similar techniques to LSTMs, we have no comments on something that has not yet been done and validated.
>
> 3. R4 commented: “While Transformer-XL has the potential to make use of word orders in the preceding sentences, it seems that this paper’s approach cannot do that, since they only take the bag-of-words from the preceding sentences. It thus seems that their bag-of-word approach is less expressive, and hence less powerful, than the simpler alternative of concatenating sentences.”
>
> Response 3: We are again surprised that R4 is so determined to completely dismiss the idea of using the bag-of-words representation just because Transformer-XL “has the potential” to make use of word orders in the preceding sentences. If the key goal is to capture document-level semantic information, discarding the word order could help better capture long-range word dependencies. As discussed in TopicRNN (Dieng et al, ICLR 2018), probabilistic topic models are a family of models that can be used to capture global semantic coherency (D. M. Blei and J. D. Lafferty. Topic models. Text mining: classification, clustering, and applications, 10(71):34, 2009), providing a powerful tool for summarizing, organizing, and navigating document collections. One basic goal of such models is to extract document-level word concurrence patterns into latent topics from a text corpus. Documents are then represented as mixtures over these latent topics. Through posterior inference, the learned topics capture the semantic coherence of the words they cluster together (Mimno et al., ACL 2011). Most topic models are “bag of words” models in that the word order is ignored, and this makes it easier for topic models to capture global semantic information compared with conventional RNN-based language models.
>
> In addition, given the learned rGBN-RNN, we can generate the sentence/paragraph given the key words of a topic, which is a unique feature of topic-guided language models.

---

> > ### Author Response · Authors · 2019-12-20
> > **Response to Review #4 (Part 2)**
> >
> > 4. R4 questioned the choice of evaluation datasets.
> >
> > Response 4: We report the perplexity results in Table 1, where the datasets are available online. We choose these datasets mainly to make direct comparison with existing topic-guided language models. Note we did consider evaluating rGBN-RNN and GBN-RNN on PTB. Unfortunately, as PTB has been pre-processed to a set of sentences, with the document boundary information discarded, we are not able to apply rGBN-RNN and GBN-RNN to PTB since they both need to know which sentences come from which documents.
> >
> > 5. R4 commented: “The inference part is not particularly self-contained.”
> >
> > Response 5: We note we have described a hybrid variational/sampling inference for rGBN-RNN in Algorithm 1 and provided the details about sampling Phi and Pi with TLASGR-MCMC in Appendix C. We also note our code is publicly available.
> >
> > 6. R4 commented: “Evaluation of the induced topic hierarchy (Figure 4) is only done through qualitative samples, and the paper does not really explain how to pick the samples (i.e. possible cherry-picking). I am not very familiar with the topic modelling literature, but it would be nice if the induced hierarchy can be evaluated quantitatively.”
> >
> > Response 6: We refuse to accept the hypothetical accusation of “possible cherry-picking.” In page 7 we have described how we visualize the topic hierarchy: “In Fig. 4, we select a large-weighted topic at the top hidden layer and move down the network to include any lower-layer topics connected to their ancestors with sufficiently large weights. Horizontal arrows link temporally related topics at the same layer, while top-down arrows link hierarchically related topics across layers.” We’d be glad to provide more analogous plots, each of which is a topic hierarchy rooted at a node of the top hidden layer (we could provide a code to automatically generate these topic hierarchy plots given the inferred Phi and Pi matrices).
> >
> > We had responded to both a public comment by Pankaj Gupta and Review # 3 about why quantitatively evaluation for the topic modeling part had not been performed. Please see these responses for details.
> >
> > We emphasize while we focus on guiding (stacked-)RNN with GBN or rGBN, the same idea can be adapted to potentially improve a Transformer based model with the help of GBN or rGBN. This extension, which is non-trivial at all due to the size of Transformer, is beyond the scope of this paper. With that said, motivated by the AC’s comments, we have been working on GBN-Transformer and rGBN-Transformer and we hope we can report comprehensive results now, but Transformer is so computationally demanding to train that our pace of progress has been limited by our current computational resource.

---

### Public Comment · ~pankaj_gupta1 · 2019-09-27
**References and Additional topic modeling evaluation**

Missing references:
[1] Hugo Larochelle and Stanislas Lauly. A neural autoregressive topic model. In NIPS 2012.
[2] Pankaj Gupta, Yatin Chaudhary, Florian Buettner, Hinrich Schütze. textTOvec: Deep Contextualized Neural Autoregressive Topic Models of Language with Distributed Compositional Prior. In ICLR 2019.

- include the neural network based topic models [1] in introduction
- As mentioned in introduction section that traditional topic models ignore word ordering. The recent work [2] addresses the issue by introducing word ordering in topic models via composite modeling of a topic model and an LSTM based language model to deal with BoW issues in topic modeling.
-  While this work focuses on improving LMs using topics, I would appreciate if you could also show quantitative results on topic modeling portion, such as topic coherence similar to Topic-RNN, TCNLM, etc.

---

> ### Author Response · Authors · 2019-11-03
> **The focus is on language model**
>
> Dear Pankaj,
>
> Thank you for suggesting both Larochelle & Lauly (2012) and your own publication in ICLR 2019.
>
> As this paper is focusing on improving a language model with a deep dynamic topic model, we think it would be unnecessary, even distracting, to evaluate topic coherence in this paper. We note there are publications, such as [3] and [4], that have evaluated the topic coherence for the topics produced by gamma belief networks (GBNs); we encourage you to check these publications for details. If we submit a paper in the future that is focused on improving a topic model with the help of a language model, we will cite your paper and add comparison, if appropriate.
>
> [3] He Zhao, Lan Du, Wray Buntine, and Mingyuan Zhou. "Dirichlet belief networks for topic structure learning." In NeurIPS 2018.
>
> [4] He Zhao, Lan Du, Wray Buntine, and Mingyuan Zhou. "Inter and intra topic structure learning with word embeddings." In International Conference on Machine Learning, pp. 5887-5896. 2018.

---

### Comment · Area_Chair1 · 2019-10-31
**How does this compare to state-of-the-art Transformer baselines?**

This paper looks quite interesting, but recent state-of-the-art results in language modeling and generation are largely based on Transformer-based models [1,2]. However, any comparison or even mention of these models seems to be conspicuously missing from this paper. I wonder: have the authors compared with any models? My suspicion is that these models are already able to capture topic to some extent, and may obviate the need for the methods proposed in this paper (but I would be happy to be proven wrong).

If not, but the authors would be interested in performing a comparison, I would suggest Transformer-XL [2], which as a model that is specifically designed to be able to capture long-distance context.

[1] Radford, Alec, et al. "Improving language understanding by generative pre-training." Preprint (2018).
[2] Dai, Zihang, et al. "Transformer-XL: Attentive language models beyond a fixed-length context." ACL (2019).

---

> ### Author Response · Authors · 2019-11-03
> **rGBN-RNN and Transformer-XL are not directly comparable**
>
> Thank you for bringing these two papers to our attention and suggesting Transformer-XL for comparison. While we are studying both papers carefully to see whether we can design additional experiments to provide meaningful comparisons, we have a number of clear reasons to explain why the proposed rGBN-RNN and Transformer-XL are not directly comparable:
>
> 1) Model size: For language modeling, the model size of Transformer-XL is one or two orders of magnitude larger than that of rGBN-RNN. For example, without considering the word embedding layers, Transformer-XL 12L and 24L have 41M and 277M parameters, respectively, while the proposed rGBN-RNN with 3 stochastic hidden layers has as few as 7.3M parameters.
>
> 2) Model construction: rGBN-RNN guides stacked-RNN, a sentence-level language model, with rGBN, a deep dynamic topic model, to construct a larger-context language model, which clearly enhances the performance of stacked-RNN. Therefore, a promising extension is to replace stacked-RNN with Transformer (which essentially consists of stacked multi-head self-attention modules), i.e., constructing a rGBN guided Transformer (rGBN-Transformer). In other words, Transformer-XL and rGBN-Transformer would be comparable. However, rGBN-Transformer is beyond the scope of the current paper and will be our future work.
>
> 3) Interpretability: In comparison to Transformer-XL, rGBN-RNN is much more interpretable and one can clearly understand its underlying mechanism to capture short-range, middle-range, and longe-range word dependencies.  For example, rGBN-RNN has the ability to learn interpretable recurrent multilayer topics from the documents, as shown in Figure 4. Besides having the ability to generate sentences from random noises, we can also generate  sentences conditioning on a single topic of a certain layer, or a combination of topics from different layers.
>
> 4) Larger-context language model: Transformer by itself is not a larger-context language model. While Transformer-XL improves Transformer to capture longer-range dependencies, it still does not respect the natural document boundary of the words. By contrast, rGBN-RNN does respect the word-sentence-document structure, using the deep dynamic topic model to guide the language model to  capture not only  the short-range local word dependencies, but also both the sequential dependencies between the document contexts of sentences, and the long-range document-level word dependencies.

---

> > ### Comment · Area_Chair1 · 2019-11-05
> > **Thank you, but still think claims need to be clarified.**
> >
> > Thank you for the clarification. In the abstract of the paper and elsewhere, it says that the proposed model "outperforms state-of-the-art larger-context language models". I would argue that state of the art larger context language models are based on Transformers, and without a comparison this claim has not been validated. Given that claims of state-of-the-art results seem to be a major selling point for the paper, I think these claims can either be toned down ("outperforms other RNN-based language models"), or an empirical comparison could be performed.

---

> > > ### Author Response · Authors · 2019-11-15
> > > **We have clarified our claims**
> > >
> > > We appreciate your comments and suggestions. Given the time constraint, we leave comprehensive comparisons between rGBN-RNN and Transformer based language models for future study. We have followed your suggestion to revise our claim to be: "the proposed model not only outperforms state-of-the-art larger-context RNN-based language models, but also..."

---

### Author Response · Authors · 2019-12-20
**Timeline of the reviewing process**

The timeline of the reviewing process for our paper is as follows:

Oct 31: The Area Chair (AC) posted public comments in OpenReview (before we could see the reviews)
Nov 3: We responded to the AC’s comments in OpenReview (the AC replied to our response on Nov 4)
Nov 5: All three reviews were released, with 8,8,8 ratings
Nov 15: We uploaded the revised paper and posted our response to the three reviewers and the AC
Dec 2: Two additional reviews were posted. Both were very negative and provided the lowest possible rating of 1
Dec 2/3: We expressed concerns to the program chairs, who kindly clarified that no additional author rebuttals are allowed before the final decision
Dec 2/3: Multiple tweets appeared in Twitter pointing to a Reddit post.
Dec 5: We wrote our response to Reviewers 4 and 5 but could not post it in OpenReview
Dec 19: The decision of "Rejection" was announced
Dec 20: We posted our response to Reviewers 4 and 5

While none of the authors were active in social media, the reviewing process of our paper that appeared out of the ordinary had caught the attention of social media, when two additional reviews, whose ratings were significantly different from the previous ones, were posted on December 2, 17 days after the rebuttal deadline and 4 days before the meta-review deadline.

We were extremely surprised by the comments of Reviewers 4 and 5. As we were not able to respond in OpenReview before the release of the final decision, we had thought about posting our response in Reddit and engaging in these discussions. An important reason that we decided not to do so, after interacting with a program chair, was that ICLR 2020 does not allow post-rebuttal interactions between the authors and the reviewers & AC, and there was no guarantee that any of the reviewers & AC had not participated or would not participate in these discussions. Thus if we did post our response and engage in these discussions before the final decision, it is possible we would have violated the rule by going around OpenReview to interact with the reviewers & AC.

Rejection is a normal part of life and it is not uncommon for reviewers from different research fields to clearly disagree with each other. The AC did raise several good concerns that we appreciated and had tried our best to respond given the time constraint. We feel that the AC could have rejected our paper based on his/her own expertise and judgment, to which we would probably have little to complain.

We hope how our paper had been reviewed could provide an example to help refine the reviewing process of ICLR and other open-reviewed machine learning conferences. For example, allowing the authors to respond to additional reviews posted after the rebuttal deadline. For all four authors, we prefer to focusing on moving our research forward, rather than getting distracted by unexpected/unwanted social media attentions that may have little to do with the quality of the submission.

---

### Decision · Program_Chairs · 2019-12-19

**Decision:**

Reject

**Comment:**

This paper was a very difficult case. All three original reviewers of the paper had never published in the area, and all of them advocated for acceptance of the paper. I, on the other hand, am an expert in the area who has published many papers, and I thought that while the paper is well-written and experimental evaluation is not incorrect, the method was perhaps less relevant given current state-of-the-art models. In addition, the somewhat non-standard evaluation was perhaps causing this fact to be masked. I asked the original reviewers to consider my comments multiple times both during the rebuttal period and after, and unfortunately none of them replied.

Because of this, I elicited two additional reviews from people I knew were experts in the field. The reviews are below. I sent the PDF to the reviewers directly, and asked them to not look at the existing reviews (or my comments) when doing their review in order to make sure that they were making a fair assessment.

Long story short, Reviewer 4 essentially agreed with my concerns and pointed out a few additional clarity issues. Reviewer 5 pointed out a number of clarity issues and was also concerned with the fact that d_j has access to all other sentences (including those following the current sentence). I know that at the end of Section 2 it is noted that at test time d_j only refers to previous sentences, but if so there is also a training-testing disconnect in model training, and it seems that this would hurt the model results.

Based on this, I have decided to favor the opinions of three experts (me and the two additional reviewers) over the opinions of the original three reviewers, and not recommend the paper for acceptance at this time. In order to improve the paper I would suggest the following (1) an acknowledgement of standard methods to incorporate context by processing sequences consisting of multiple sentences simultaneously, (2) a more thorough comparison with state-of-the-art models that consider cross-sentential context on standard datasets such as WikiText or PTB. I would encourage the authors to consider this as they revise their paper.

Finally, I would like to apologize to the authors that they did not get a chance to reply to the second set of reviews. As I noted above, I did try to make my best effort to encourage discussion during the rebuttal period.